# Multi-block Min-max Bilevel Optimization with Applications in Multi-task Deep AUC Maximization

**Quanqi Hu**
Department of Computer Science
Texas A&M University
College Station, TX 77843
`quanqi-hu@tamu.edu`

**Yongjian Zhong**
Department of Computer Science
University of Iowa
Iowa City, IA 52242
`yongjian-zhong@uiowa.edu`

**Tianbao Yang**
Department of Computer Science
Texas A&M University
College Station, TX 77843
`tianbao-yang@tamu.edu`

## Abstract

In this paper, we study multi-block min-max bilevel optimization problems, where the upper level is non-convex strongly-concave minimax objective and the lower level is a strongly convex objective, and there are multiple blocks of dual variables and lower level problems. Due to the intertwined multi-block min-max bilevel structure, the computational cost at each iteration could be prohibitively high, especially with a large number of blocks. To tackle this challenge, we present two single-loop randomized stochastic algorithms, which require updates for only a constant number of blocks at each iteration. Under some mild assumptions on the problem, we establish their sample complexity of $\mathcal{O}(1/\epsilon^4)$ for finding an $\epsilon$-stationary point. This matches the optimal complexity for solving stochastic non-convex optimization under a general unbiased stochastic oracle model. Moreover, we provide two applications of the proposed method in multi-task deep AUC (area under ROC curve) maximization and multi-task deep partial AUC maximization. Experimental results validate our theory and demonstrate the effectiveness of our method on problems with hundreds of tasks.

## 1 Introduction

We consider multi-block min-max bilevel optimization problem of the following formulation

$$
\min_{\mathbf{x}\in\mathbb{R}^{d_x}} \max_{\boldsymbol{\alpha}\in\mathcal{A}^m} F(\mathbf{x},\boldsymbol{\alpha}) := \frac{1}{m}\sum_{i=1}^{m}\left\{f_i(\mathbf{x},\alpha_i,\mathbf{y}_i(\mathbf{x})) := \mathbb{E}_{\xi\in\mathcal{P}_i}[f_i(\mathbf{x},\alpha_i,\mathbf{y}_i(\mathbf{x});\xi)]\right\} \quad \text{(upper)}
$$
$$
\text{s.t. } \mathbf{y}_i(\mathbf{x}) = \arg\min_{\mathbf{y}_i\in\mathbb{R}^{d_y}} g_i(\mathbf{x},\mathbf{y}_i) := \mathbb{E}_{\zeta\in\mathcal{Q}_i}[g_i(\mathbf{x},\mathbf{y}_i;\zeta)], \quad \text{for } i=1,2,\dots,m. \quad \text{(lower)}
$$

(1)

where $f_i$ and $g_i$ are smooth functions and $\mathcal{A}\subset\mathbb{R}^{d_\alpha}$ is a convex set. In particular, in this paper we assume that for each $i\in\{1,\dots,m\}$, $f_i(\mathbf{x},\alpha_i,\mathbf{y}_i)$ is strongly concave in the dual variable $\alpha_i$ but can be nonconvex in the primal variable $\mathbf{x}$, and $g_i(\mathbf{x},\mathbf{y}_i)$ is strongly convex in $\mathbf{y}_i$. The *upper problem* $\min_{\mathbf{x}\in\mathbb{R}^{d_x}}\max_{\boldsymbol{\alpha}\in\mathcal{A}^m} F(\mathbf{x},\boldsymbol{\alpha})$ is a min-max optimization problem where the *lower problems* $\{\mathbf{y}_i(\mathbf{x}) = \arg\min_{\mathbf{y}_i\in\mathbb{R}^{d_y}} g_i(\mathbf{x},\mathbf{y}_i)\}_{i=1}^{m}$ are involved as variables. For each *block i*, the *upper-level objective* $f_i(\mathbf{x},\alpha_i,\mathbf{y}_i)$ and the *lower-level objective* $g_i(\mathbf{x},\mathbf{y}_i)$ depend only on its corresponding block

36th Conference on Neural Information Processing Systems (NeurIPS 2022).

of variables $\boldsymbol{\alpha}$ and $\mathbf{y}$, i.e. $\alpha_i$ and $\mathbf{y}_i$. This problem has important applications in machine learning, e.g., multi-task deep AUC maximization as presented in section 3.

Tackling problem (1) is challenging as it involves solving a min-max problem with coupled multiple minimization problems simultaneously. The naive way for solving it is to do multiple gradient ascents and descents for $\boldsymbol{\alpha}$ and $\mathbf{y}$, respectively, to ensure a good estimation of the gradient for updating $\mathbf{x}$. However, this approach has two major drawbacks. As the algorithm involves a double loop structure, it can be computationally expensive and give suboptimal theoretical complexity. On the other hand, the multi-block structure requires data sampling from distributions $\mathcal{P}_i, \mathcal{Q}_i$ for all blocks, which may lead to an impractical demand for memory.

## 1.1 Related work

**Min-max Bilevel optimization.** To the best of our knowledge, the only existing work that provides a stochastic algorithm with provable convergence guarantee on min-max bilevel problems is [9]. They propose a single loop bi-time scale stochastic algorithm based on gradient descent ascent, and prove that it converges to an $\epsilon$-stationary point with an oracle complexity of $\mathcal{O}(\epsilon^{-5})$. Nevertheless, this convergence result is established for a special case where $f(\mathbf{x}, \cdot, \mathbf{y})$ is a linear function.

**Stochastic Nonconvex Strongly Concave Min-max Problems.** The considered problem is also closely related to non-convex strongly concave min-max problems, which have been studied extensively recently. To the best of our knowledge, [30] establishes the first result on non-smooth nonconvex concave min-max problems. They prove a convergence to nearly stationary point of the primal objective function with an oracle complexity in the order of $\mathcal{O}(\epsilon^{-4})$ for non-convex strongly concave min-max problems with a certain special structure. The same order of oracle complexity is achieved in [36] without relying on any special structure. These two works use two-loop algorithms. There are some studies focusing on single-loop algorithms. [25] analyzes a single-loop stochastic gradient descent ascent (SGDA) method for smooth nonconvex strongly concave problem, which achieves $\mathcal{O}(\epsilon^{-4})$ complexity but with a large mini-batch size. In [12], the same order of complexity is achieved without large mini-batch by employing the stochastic moving average estimator. Some recent works improve the sample complexity to $\mathcal{O}(\epsilon^{-3})$ under the Lipschitz continuous oracle model for the stochastic gradient using less practical variance reduction techniques [16, 28, 31]. [41] establishes lower complexity bounds for non-concex strongly concave min-max problem under both general and finite-sum setting and proposes accelerated algorithms that nearly match the lower bounds.

**Stochastic Nonconvex Bilevel optimization.** The considered problem belongs to a general family of non-convex bilevel optimization problems. Non-asymptotic convergence results for nonconvex stochastic bilevel optimization (SBO) with a strongly convex lower problem has been established in several recent studies [4, 8, 12, 14, 19]. As the one who gives the first results for this problem, [8] proposes a double-loop algorithm with $\mathcal{O}(\epsilon^{-6})$ oracle complexity for finding an $\epsilon$-stationary point of the objective function. [19] improves the complexity order to $\mathcal{O}(\epsilon^{-4})$, but suffers from a large mini-batch size. [14] proposes a single-loop algorithm with two time-scale updates that achieves an oracle complexity of $\widetilde{\mathcal{O}}(\epsilon^{-5})$. Recently, [12] improves the oracle complexity to the state-of-the-art oracle complexity $\widetilde{\mathcal{O}}(\epsilon^{-4})$ by proposing a single-loop algorithm based on a moving-average estimator. [5] presents a new analysis for (double-loop) SGD-type updates showing that an improved sample complexity $\mathcal{O}(\epsilon^{-4})$ can be achieved. There are studies that further improve the complexity to $\mathcal{O}(\epsilon^{-3})$ by leveraging the Lipschitz continuous conditions of stochastic oracles [4, 10, 21]. [24] considers bilevel optimization under distributed setting and proposed algorithms achieving state-of-the-art complexities. However, none of these works tackle multi-block min-max bilevel optimization problems directly.

**Multi-block Bilevel Optimization.** There are some recent studies considering bilevel optimization with multi-block structure. [10] extends their single-block bilevel optimization algorithm to multi-block structure. Their algorithm requires two independently sampled block batches and for all sampled blocks, each variable needs update using variance reduction technique STORM [6]. For unsampled blocks, an update involving constant factor multiplication is also required. Under Lipschitz continuous conditions on stochastic oracles, the complexity is no worse than $\mathcal{O}(m/\epsilon^3)$ with $m$ blocks. A more recent work [29] considers top-K NDCG optimization, which is formulated as a compositional bilevel optimization with multi-block structure. Their method simplifies the updates by sampling only one block batch in each iteration and requires updates only for the sampled blocks. Their method achieves complexity of $\mathcal{O}(m/\epsilon^4)$. We use a similar approach as the latter work for estimating the hessian inverse in a block-wise manner. However, this paper differs from [29] in that we tackle a more general multi-block min-max bilevel problems without assuming a particular form of the objective.

## 1.2 Our Contributions

In Section 2.1, we present two simple single loop single timescale stochastic methods with randomized block-sampling for solving a general form of multi-block min-max bilevel optimization problem under the nonconvex strongly concave (upper) strongly convex (lower) setting. Both methods employ SGD for updating selected $\mathbf{y}_i$ for their corresponding lower-level problems, employs SGA for updating the selected $\alpha_i$, and employs a momentum update for the primal variable $\mathbf{x}$ based on the sampled $\alpha_i, \mathbf{y}_i$. Then we show in Section 2.2, theoretically, that they converge to $\epsilon$-stationary point with complexity $\mathcal{O}(\epsilon^{-4})$ under a general unbiased stochastic oracle model. Our result for the single-block setting matches the lower bound for solving smooth, potentially nonconvex optimization through queries to an unbiased stochastic gradient oracle under a bounded variance condition [1]. Finally, in section 3 we present two applications of multi-block min-max bilevel optimization in deep AUC maximization: multi-task deep AUC maximization and multi-task deep partial AUC maximization. In section 4, empirical results show the effectiveness of the proposed methods.

# 2 Proposed Algorithms and convergence Analysis

**Notations.** Let $\|\cdot\|$ denote the Euclidean norm of a vector or the spectral norm of a matrix. For a twice differentiable function $f : X \times Y \to \mathbb{R}$, $\nabla_x f(x, y)$ (resp. $\nabla_y f(x, y)$) denotes its partial gradient taken w.r.t $x$ (resp. $y$), and $\nabla^2_{xy} f(x, y)$ (resp. $\nabla^2_{yy} f(x, y)$) denotes the Jacobian of $\nabla_x f(x, y)$ w.r.t x (resp. $\nabla_y f(x, y)$ w.r.t y). We let $f(\cdot; \mathcal{B})$ represent the unbiased stochastic oracle of $f(\cdot)$ with a sample batch $\mathcal{B}$ as the input. The unbiased stochastic oracle is said to have bounded variance $\sigma^2$ if $\mathbb{E}[\|f(\cdot; \mathcal{B}) - f(\cdot)\|^2] \leq \sigma^2$. A mapping $f : X \to \mathbb{R}$ is $C$-Lipschitz continuous if $\|f(x) - f(x')\| \leq C\|x - x'\| \; \forall x, x' \in X$. Function $f$ is $L$-smooth if its gradient $\nabla f(\cdot)$ is $L$-Lipschitz continuous. A function $g : X \to \mathbb{R}$ is $\lambda$-strongly convex if $\forall x, x' \in X$, $g(x) \geq g(x') + \nabla g(x')^T (x - x') + \frac{\lambda}{2}\|x - x'\|^2$. A function $g : X \to \mathbb{R}$ is $\lambda$-strongly concave if $-g(x)$ is $\lambda$-strongly convex. Let $\Pi_{\mathcal{A}}$ denote a projection function onto a convex set $\mathcal{A}$. For notation simplicity, we use $\mathcal{S}$ to denote the set of all block indices, *i.e.* $\mathcal{S} = \{1, \ldots, m\}$.
We state the definition of $\epsilon$-stationary point as following.

**Definition 2.1.** Consider a differentiable function $F(\mathbf{x})$, a point $\mathbf{x}$ is called $\epsilon$-stationary if $\|\nabla F(\mathbf{x})\| \leq \epsilon$. A stochastic algorithm is said to achieve an $\epsilon$-stationary point if $\mathbb{E}[\|\nabla F(\bar{\mathbf{x}}_t)\|] \leq \epsilon$, where $\bar{\mathbf{x}}_t$ is the algorithm output at the $t$-th iteration and the expectation is taken over the randomness of the algorithm until the iteration $t$.

**Assumptions.** Before presenting our algorithm, we make the following well-behaving assumptions.

**Assumption 2.2.** For functions $f_i$ and $g_i$, we assume that the following conditions hold for all $i \in \mathcal{S}$

- $f_i(\mathbf{x}, \alpha_i, \mathbf{y}_i)$ is $\mu_f$-strongly concave in terms of $\alpha_i$. $g_i(x, \mathbf{y}_i)$ is $\mu_g$-strongly convex in terms of $\mathbf{y}_i$.

- $f_i$ is $C_f$-Lipschitz continuous in terms of both $\mathbf{x}$ and $\mathbf{y}_i$, and $f_i, g_i$ are $L_f, L_g$-smooth respectively.

- $\|\nabla^2_{xy} g_i(\mathbf{x}, \mathbf{y}_i)\|^2 \leq C^2_{gxy}$, $\nabla^2_{yy} g_i(\mathbf{x}, \mathbf{y}_i; \zeta) \succeq \mu_g I$.

- $\nabla^2_{xy} g_i(\mathbf{x}, \mathbf{y}_i), \nabla^2_{yy} g_i(\mathbf{x}, \mathbf{y}_i)$ are $L_{gxy}, L_{gyy}$-Lipschitz continuous respectively.

We remark that the Lipschitz continuity condition of $f_i$ in terms of $\mathbf{x}$ can be removed when there is only one block. Other Lipschitz continuity conditions are stadnard in bilevel optimization literature. Moreover, the gradients of functions $f_i$ and $g_i$ can only be accessed through unbiased oracles with bounded variance.

**Assumption 2.3.** The unbiased stochastic oracles $\nabla_x f_i(\mathbf{x}, \alpha_i, \mathbf{y}_i; \mathcal{B})$, $\nabla_\alpha f_i(\mathbf{x}, \alpha_i, \mathbf{y}_i; \mathcal{B})$, $\nabla_y f_i(\mathbf{x}, \alpha_i, \mathbf{y}_i; \mathcal{B}), \nabla_y g_i(\mathbf{x}, \mathbf{y}_i; \mathcal{B}), \nabla^2_{xy} g_i(\mathbf{x}, \mathbf{y}_i; \mathcal{B}), \nabla^2_{yy} g_i(\mathbf{x}, \mathbf{y}_i; \mathcal{B})$ have variances bounded by $\frac{\sigma^2}{|\mathcal{B}|}$ for all $i \in \mathcal{S}$, where $|\mathcal{B}|$ denotes the size of the sampled batch $\mathcal{B}$.

These assumptions are similar to those made in many existing works for SBO [4, 8, 14, 19].

**Moving average gradient estimator.** Algorithms based on moving average estimators have achieved the state-of-the-art oracle complexity in both min-max and bilevel optimizations [11]. Here we give a brief introduction to the moving average estimator. For solving a nonconvex minimization problem $\min_{\mathbf{x} \in \mathbb{R}^d} F(\mathbf{x})$ through an unbiased oracle $\mathcal{O}_F(\mathbf{x})$, i.e. $\mathbb{E}[\mathcal{O}_F(\mathbf{x})] = \nabla F(\mathbf{x})$, the stochastic momentum method (stochastic heavy-ball method) that employs moving average updates is given by

$$\mathbf{v}_{t+1} = (1 - \beta)\mathbf{v}_t + \beta \mathcal{O}_F(\mathbf{x}_t), \qquad \mathbf{x}_{t+1} = \mathbf{x}_t - \eta \mathbf{v}_{t+1},$$

---

**Algorithm 1** A Stochastic Algorithm for Multi-block Min-max Bilevel Optimization (v1)

---

**Require:** $\alpha^0, \mathbf{y}^0, H^0, \mathbf{z}_0, \mathbf{x}_0$

1: **for** $t = 0, 1, \ldots, T$ **do**
2:     Sample tasks $I_t \subset \mathcal{S}$. Sample data batches $\mathcal{B}_i^t \subset \mathcal{P}_i$, $\tilde{\mathcal{B}}_i^t \subset \mathcal{Q}_i$ of batch size $B$ for each $i \in I_t$.
3:     **for** sampled blocks $i \in I_t$ **do**
4:         $\alpha_i^{t+1} = \Pi_{\mathcal{A}}[\alpha_i^t + \eta_1 \nabla_\alpha f_i(\mathbf{x}_t, \alpha_i^t, \mathbf{y}_i^t; \mathcal{B}_i^t)]$
5:         $\mathbf{y}_i^{t+1} = \mathbf{y}_i^t - \eta_2 \nabla_y g_i(\mathbf{x}_t, \mathbf{y}_i^t; \tilde{\mathcal{B}}_i^t)$
6:     **end for**
7:     Update estimator $H^{t+1}$ of $[\nabla_{yy}^2 g_i(\mathbf{x}_t, \mathbf{y}_i^t)]^{-1}$ by (2)
8:     Update gradient estimator $\Delta^{t+1}$ of $\nabla_x F(\mathbf{x}_t)$ by (3)
9:     $\mathbf{z}_{t+1} = (1 - \beta_0)\mathbf{z}_t + \beta_0 \Delta^{t+1}$
10:    $\mathbf{x}_{t+1} = \mathbf{x}_t - \eta_0 \mathbf{z}_{t+1}$
11: **end for**

---

where $\beta$ and $\eta$ are momentum parameter and learning rate, respectively. As a moving average of the historical gradient estimator, the sequence of $\mathbf{v}_{t+1}$ could achieve an effect of variance diminishing across a long run ([32]).

## 2.1 The Proposed Algorithms

First, we propose a simple single loop stochastic algorithm 1 to solve the multi-block min-max bilevel optimization problem. At the beginning of each iteration, we first sample a set of blocks $I_t$ and data batches $\mathcal{B}_i^t, \tilde{\mathcal{B}}_i^t$ for each selected block $i \in I_t$. Then we update estimators of $\alpha_i(\mathbf{x}_t)$ and $\mathbf{y}_i(\mathbf{x}_t)$ for all selected blocks $i \in I_t$ using one step of SGA and SGD. Then, we compute an estimator of the hessian inverse $H^{t+1}$ of the lower-level objective and compute a gradient estimator $\Delta^{t+1}$ of the upper-level objective. Finally, we compute the moving average estimator $\mathbf{z}_{t+1}$ of $\nabla F(\mathbf{x}_t)$ and update $\mathbf{x}_{t+1}$. Note that the design of our algorithm on the min-max bilevel optimization part is inspired by [11], and it is similar to their momentum-based algorithms PDSM (for min-max problem) and SMB (for bilevel problem) in their paper. In fact, if we set the number of blocks to be one and remove $\mathbf{y}$ and the lower level problems, then Algorithm 1 is the same as PDSM. Similarly, if the number of blocks is one and the dual variables in the upper level problem are removed, then the proposed algorithm becomes similar as SMB except for the hessian inverse update with reasons explain shortly. In other words, our proposed method is a generalized form of momentum-based algorithm for min-max and bilevel optimization problems. Additionally, Algorithm 1 only updates $O(1)$ blocks of dual variables $\alpha_i$ and the variables $\mathbf{y}_i$ of the lower-level problems. These make the analysis of Algorithm 1 much more involved.

To further understand Algorithm 1, we first define the objective function $F(\mathbf{x}) := \frac{1}{m} \sum_{i \in \mathcal{S}} f_i(\mathbf{x}, \alpha_i(\mathbf{x}), \mathbf{y}_i(\mathbf{x}))$, where $\mathbf{y}_i(\mathbf{x}) = \arg\min_{\mathbf{y}_i} g_i(\mathbf{x}, \mathbf{y}_i)$ and $\alpha_i(\mathbf{x}) := \arg\max_{\alpha_i} f_i(\mathbf{x}, \alpha_i, \mathbf{y}_i(\mathbf{x}))$, so that the Problem (1) can be rewritten as $\min_{\mathbf{x}} F(\mathbf{x})$. The updates for $\alpha_i^{t+1}$'s and $\mathbf{y}_i^{t+1}$'s are intuitive since the gradient estimations of $\nabla_\alpha f_i(\mathbf{x}_t, \alpha_i^t, \mathbf{y}_i^t)$ and $\nabla_y g_i(\mathbf{x}_t, \mathbf{y}_i^t)$ are directly available from the unbiased stochastic oracles. However, since functions $\mathbf{y}_i(\mathbf{x})$ and $\alpha_i(\mathbf{x})$ are implicit, estimating the gradient $\nabla F(\mathbf{x})$ is difficult. In fact, one may apply the corollary of Theorem 1 in [3] to get:

$$\nabla F(\mathbf{x}) = \frac{1}{m} \sum_{i \in \mathcal{S}} \left( \nabla_x f_i(\mathbf{x}, \alpha_i(\mathbf{x}), \mathbf{y}_i(\mathbf{x})) + \nabla \mathbf{y}_i(\mathbf{x}) \nabla_y f_i(\mathbf{x}, \alpha_i(\mathbf{x}), \mathbf{y}_i(\mathbf{x})) \right).$$

A standard approach in bilevel optimization literature [8] for computing $\nabla \mathbf{y}_i(\mathbf{x})$ is to derive $\nabla \mathbf{y}_i(\mathbf{x}) = -\nabla_{xy}^2 g_i(\mathbf{x}, \mathbf{y}_i(\mathbf{x}))[\nabla_{yy}^2 g_i(\mathbf{x}, \mathbf{y}_i(\mathbf{x}))]^{-1}$ from the optimality condition of $\mathbf{y}_i(\mathbf{x})$. Therefore, the gradient we are looking for is given by

$$\nabla F(\mathbf{x}) = \frac{1}{m} \sum_{i \in \mathcal{S}} \nabla_x f_i(\mathbf{x}, \alpha_i(\mathbf{x}), \mathbf{y}_i(\mathbf{x})) - \nabla_{xy}^2 g_i(\mathbf{x}, \mathbf{y}_i(\mathbf{x}))[\nabla_{yy}^2 g_i(\mathbf{x}, \mathbf{y}_i(\mathbf{x}))]^{-1} \nabla_y f_i(\mathbf{x}, \alpha_i(\mathbf{x}), \mathbf{y}_i(\mathbf{x})).$$

All components in this gradient can be easily obtained from unbiased stochastic oracles except for the inverse of hessian $[\nabla_{yy}^2 g_i(\mathbf{x}, \mathbf{y}_i(\mathbf{x}))]^{-1}$ for all blocks. This could be problematic in the sense of theory and practical implementation. For practical implementation, we do not want to update the hessian inverse estimators for all blocks, which is prohibitive when the number of

blocks is large. A common approach used in the literature of SBO is to use Neumann series [8] $H_i^{t+1} = \frac{k_t}{C_{g_{yy}}} \prod_{j=1}^{q} \left( I - \frac{1}{C_{g_{yy}}} \nabla_{yy}^2 g_i(\mathbf{x}_t, \mathbf{y}_i^t; \xi_i^t) \right)$, where $q$ is chosen from $\{1, \dots, k_t\}$ randomly and $k_t$ is the number of samples $\{\xi_i^t\}_{i=1}^{k_t}$ for estimating the hessian inverse. This estimator is a biased one and its error w.r.t to $\nabla_{yy}^2 g_i(\mathbf{x}_t, \mathbf{y}_i^t)^{-1}$ is controlled by the number of samples $k_t$ [8]. However, it is problematic to employ the above estimator for only the sampled blocks $i \in I_t$ because the error for those not sampled cannot be controlled. To address this issue, we use a different approach for estimating the hessian inverse by only updating the estimators for those sampled blocks [29]. The idea is to maintain a momentum term $s_i^{t+1}$ for each block that stores historical information on the hessian estimator $\nabla_{yy}^2 g_i(\mathbf{x}_t, \mathbf{y}_i^t; \tilde{\mathcal{B}}_i^t)$. And the hessian inverse is approximated by directly computing the inverse of $s_i^{t+1}$, i.e., for sampled $i \in I_t$,

$$s_i^{t+1} = (1 - \beta_1)s_i^t + \beta_1 \nabla_{yy}^2 g_i(\mathbf{x}_t, \mathbf{y}_i^t; \tilde{\mathcal{B}}_i^t), \quad H_i^{t+1} = [s_i^{t+1}]^{-1} \tag{2}$$

In terms of theoretical analysis, we are not bounding the individual error $H_i^{t+1} - \nabla_{yy}^2 g_i(\mathbf{x}_t, \mathbf{y}_i^t)^{-1}$ for all blocks, but the cumulative error for all blocks across all iterations. This is exhibited in Lemma 2.6. As the conclusion of the above discussion, the gradient estimator of $\nabla F(\mathbf{w}_t)$ is given by

$$\Delta^{t+1} = \frac{1}{|I_t|} \sum_{i \in I_t} \left\{ \nabla_x f_i(\mathbf{x}_t, \alpha_i^t, \mathbf{y}_i^t; \mathcal{B}_i^t) - \nabla_{xy} g_i(\mathbf{x}_t, \mathbf{y}_i^t; \tilde{\mathcal{B}}_i^t) H_i^t \nabla_y f_i(\mathbf{x}_t, \alpha_i^t, \mathbf{y}_i^t; \mathcal{B}_i^t) \right\}. \tag{3}$$

We maintain a moving average estimator $\mathbf{z}_{t+1}$ for $\Delta^{t+1}$ and finally update $\mathbf{x}_{t+1}$ using $\mathbf{z}_{t+1}$. The detailed steps are presented in Algorithm 1.

Nevertheless, such method is not suitable for problems with a high dimensionality of $\mathbf{y}_i$, since computing the Hessian inverse could be computationally expensive. To this end, we propose the second method, Algorithm 2, for problems with high dimensionality of $\mathbf{y}_i$. The main idea is to treat $[\nabla_{yy}^2 g_i(\mathbf{x}, \mathbf{y}_i)]^{-1} \nabla_y f_i(\mathbf{x}, \alpha_i, \mathbf{y}_i)$ as the solution to a quadratic function minimization problem. As a result, $[\nabla_{yy}^2 g_i(\mathbf{x}, \mathbf{y}_i)]^{-1} \nabla_y f_i(\mathbf{x}, \alpha_i, \mathbf{y}_i)$ can be approximated by SGD. Such method for Hassian inverse computation has been studied for solving single-block bilevel optimization problems in some previous works [7, 23]. However, none of them has applied this method in multi-block scenario.

Define quadratic function $\gamma_i$ and its minimum point as following

$$\mathbf{v}_i(\mathbf{x}, \alpha_i, \mathbf{y}_i) := \arg\min_{v \in \mathbb{R}^{d_y}} \gamma_i(v, \mathbf{x}, \alpha_i, \mathbf{y}_i) := \frac{1}{2} v^T \nabla_{yy}^2 g_i(\mathbf{x}, \mathbf{y}_i) v - v^T \nabla_y f_i(\mathbf{x}, \alpha_i, \mathbf{y}_i)$$

Then we have the gradient $\nabla_v \gamma_i(v, \mathbf{x}, \alpha_i, \mathbf{y}_i) = \nabla_{yy}^2 g_i(\mathbf{x}, \mathbf{y}_i) v - \nabla_y f_i(\mathbf{x}, \alpha_i, \mathbf{y}_i)$, which implies that the unique solution is given by $\mathbf{v}_i(\mathbf{x}, \alpha_i, \mathbf{y}_i) = [\nabla_{yy}^2 g_i(\mathbf{x}, \mathbf{y}_i)]^{-1} \nabla_y f_i(\mathbf{x}, \alpha_i, \mathbf{y}_i)$. Note that due to the smoothness of $g$ and Lipschitz continuity of $f$ with respect to $\mathbf{y}$ in Assumption 2.2, one may define constant $\Gamma = \frac{C_f}{\mu_g}$ so that $\|\mathbf{v}_i(\mathbf{x}, \alpha_i, \mathbf{y}_i)\|^2 \leq \Gamma^2$. Considering the updates in Algorithm 2, we have $\|\mathbf{v}_i^t\|^2 \leq \Gamma^2$ for all $i, t$. Define the stochastic estimator $\nabla_v \gamma_i(v, \mathbf{x}, \alpha_i, \mathbf{y}_i; \mathcal{B}_i, \tilde{\mathcal{B}}_i) := \nabla_{yy}^2 g_i(\mathbf{x}, \mathbf{y}_i; \tilde{\mathcal{B}}_i) v - \nabla_y f_i(\mathbf{x}, \alpha_i, \mathbf{y}_i; \mathcal{B}_i)$, then it has bounded variance, of which the proof is deferred to Appendix B. Here we enlarge the value of $\sigma$ so that $\mathbb{E}_{\mathcal{B}_i^t}[\|\nabla_v \gamma_i(\mathbf{v}_i^t, \mathbf{x}_t, \mathbf{y}_i^t; \mathcal{B}_i^t, \tilde{\mathcal{B}}_i^t) - \nabla_v \gamma_i(\mathbf{v}_i^t, \mathbf{x}_t, \mathbf{y}_i^t)\|^2] \leq \frac{\sigma^2}{B}$. It is worth to note that the projection in the updates of $\mathbf{v}_i^{t+1}$ is necessary in order to bound the variance of $\nabla_v \gamma_i(\mathbf{v}_i^t, \mathbf{x}_t, \mathbf{y}_i^t; \mathcal{B}_i^t, \tilde{\mathcal{B}}_i^t)$. Instead of taking projection, previous works [7, 23] treat the variance boundedness as an assumption, which is not guaranteed without using projection.

## 2.2 Convergence Analysis

In this section, we present brief convergence analysis of Algorithm 1 and Algorithm 2. The detailed theorems and proofs are deferred to the appendix.

### 2.2.1 Convergence analysis of Algorithm 1

The key point of this analysis is the gap between the true gradient $\nabla F(\mathbf{x}_t)$ and its estimator $\mathbf{z}^{t+1}$. To this end, we define

$$\nabla F(\mathbf{x}_t, \boldsymbol{\alpha}^t, \mathbf{y}^t) := \frac{1}{m} \sum_{i \in \mathcal{S}} \left\{ \nabla_x f_i(\mathbf{x}_t, \alpha_i^t, \mathbf{y}_i^t) - \nabla_{xy}^2 g_i(\mathbf{x}_t, \mathbf{y}_i^t) \mathbb{E}_t[H_i^t] \nabla_y f_i(\mathbf{x}_t, \alpha_i^t, \mathbf{y}_i^t) \right\}.$$

---

**Algorithm 2** A Stochastic Algorithm for Multi-block Min-max Bilevel Optimization (v2)

---

**Require:** $\alpha^0, \mathbf{y}^0, \mathbf{v}^0, \mathbf{z}_0, \mathbf{x}_0$

1: **for** $t = 0, 1, \ldots, T$ **do**
2:     Sample tasks $I_t \in \mathcal{S}$. Sample data batches $\mathcal{B}_i^t \subset \mathcal{P}_i, \tilde{\mathcal{B}}_i^t \subset \mathcal{Q}_i$ of batch size $B$ for each $i \in I_t$.
3:     **for** sampled blocks $i \in I_t$ **do**
4:         $\alpha_i^{t+1} = \Pi_{\mathcal{A}}[\alpha_i^t + \eta_1 \nabla_\alpha f_i(\mathbf{x}_t, \alpha_i^t, \mathbf{y}_i^t; \mathcal{B}_i^t)]$
5:         $\mathbf{y}_i^{t+1} = \mathbf{y}_i^t - \eta_2 \nabla_y g_i(\mathbf{x}_t, \mathbf{y}_i^t; \tilde{\mathcal{B}}_i^t)$
6:         $\mathbf{v}_i^{t+1} = \Pi_\Gamma \left[ \mathbf{v}_i^t - \eta_3 \left[ \nabla_{yy}^2 g_i(\mathbf{x}_t, \mathbf{y}_i^t; \tilde{\mathcal{B}}_i^t) \mathbf{v}_i^t - \nabla_y f_i(\mathbf{x}_t, \alpha_i^t, \mathbf{y}_i^t; \mathcal{B}_i^t) \right] \right]$
7:     **end for**
8:     Update gradient estimator $\Delta^{t+1} = \frac{1}{|I_t|} \sum_{i \in I_t} \left[ \nabla_x f_i(\mathbf{x}_t, \alpha_i^t, \mathbf{y}_i^t; \mathcal{B}_i^t) - \nabla_{xy}^2 g_i(\mathbf{x}_t, \mathbf{y}_i^t; \tilde{\mathcal{B}}_i^t) \mathbf{v}_i^t \right]$
9:     $\mathbf{z}_{t+1} = (1 - \beta_0)\mathbf{z}_t + \beta_0 \Delta^{t+1}$
10:    $\mathbf{x}_{t+1} = \mathbf{x}_t - \eta_0 \mathbf{z}_{t+1}$
11: **end for**

---

One may notice that the estimator $\Delta^{t+1}$ is in fact approximating $\nabla_x F(\mathbf{x}_t, \alpha^t, \mathbf{y}^t)$ instead of $\nabla_x F(\mathbf{x}_t)$. We exploit the moving average formulation of $\mathbf{z}^{t+1}$ and decompose the gap into two parts, $\|\Delta^{t+1} - \nabla_x F(\mathbf{x}_t, \alpha^t, \mathbf{y}^t)\|^2$ and $\|\nabla_x F(\mathbf{x}_t, \alpha^t, \mathbf{y}^t) - \nabla_x F(\mathbf{x}_t)\|^2$. These two gaps are determined by how well $\mathbf{y}^t$, $\alpha^t$ and $H_i^t$ approximate $\mathbf{y}(\mathbf{x}_t)$, $\alpha(\mathbf{x}_t)$ and $[\nabla_{yy}^2 g_i(\mathbf{x}_t, \mathbf{y}_i^t)]^{-1}$, respectively. In other words, we aim to bound the following three errors, $\|\mathbf{y}(\mathbf{x}_t) - \mathbf{y}^t\|^2 =: \delta_{y,t}$, $\|\alpha(\mathbf{x}_t) - \alpha^t\|^2 := \delta_{\alpha,t}$ and $\|[\nabla_{yy}^2 g_i(\mathbf{x}_t, \mathbf{y}_i^t)]^{-1} - H_i^t\|^2$.

We first bound the variance $\mathbb{E}[\delta_{y,t}]$ by proving the following lemma.

**Lemma 2.4.** *Consider the updates for $\mathbf{y}^t$ in Algorithm 1, under Assumption 2.2 and 2.3, with $\eta_2 \leq \min\{\frac{\mu_g}{L_g^2}, \frac{2m}{|I_t|\mu_g}\}$ we have*

$$\sum_{t=0}^T \mathbb{E}[\delta_{y,t}] \leq \frac{2m}{|I_t|\eta_2\mu_g}\delta_{y,0} + \frac{4m\eta_2 T\sigma^2}{\mu_g B} + \frac{8m^3 C_y^2 \eta_0^2}{|I_t|^2 \eta_2^2 \mu_g^2} \sum_{t=0}^{T-1} \mathbb{E}[\|\mathbf{z}_{t+1}\|^2]$$

One may also bound the second variance $\mathbb{E}[\delta_{\alpha,t}]$ based on the previous variance $\mathbb{E}[\delta_{y,t}]$ following a similar strategy.

**Lemma 2.5.** *Consider the updates for $\alpha^t$ in Algorithm 1, under Assumption 2.2, 2.3, with $\eta_1 \leq \min\left\{\frac{\mu_f}{L_f^2}, \frac{1}{\mu_f}, \frac{4m}{\mu_f |I_t|}\right\}$, we have*

$$\sum_{t=0}^T \mathbb{E}_t[\delta_{\alpha,t}] \leq \frac{4m}{\eta_1\mu_f|I_t|}\delta_{\alpha,0} + \frac{24L_f^2}{\mu_f^2} \sum_{t=0}^{T-1} \mathbb{E}[\delta_{y,t}] + \frac{8m\mu_f\eta_1\sigma^2 T}{B} + \frac{32m^3 C_\alpha^2 \eta_0^2}{\eta_1^2 \mu_f^2 |I_t|^2} \sum_{t=0}^{T-1} \mathbb{E}[\|\mathbf{z}_{t+1}\|^2].$$

Due to the lower bound assumption of $\nabla_{yy}^2 g_i(\mathbf{x}, \mathbf{y}_i; \zeta)$ in Assumption 2.2, the error in Hessian approximation can be bounded by bounding $\delta_{gyy,t} := \sum_{i \in \mathcal{S}} \|s_i^t - \nabla_{yy}^2 g_i(\mathbf{x}_t, \mathbf{y}_i(\mathbf{x}_t))\|^2$. We prove the following lemma.

**Lemma 2.6.** *Under Assumption 2.2, 2.3, considering the momentum method (2) for the update of hessian, with $\beta_1 \leq 1$ we have*

$$\sum_{t=0}^T \mathbb{E}[\delta_{gyy,t}] \leq \frac{4m\delta_{gyy,0}}{|I_t|\beta_1} + 32L_{gyy}^2 \sum_{t=0}^{T-1} \mathbb{E}[\delta_{y,t}] + \frac{8m\beta_1 T\sigma^2}{B} + \frac{32m^3 L_{gyy}^2(1 + C_y^2)\eta_0^2}{|I_t|^2\beta_1^2} \sum_{t=0}^{T-1} \mathbb{E}[\|\mathbf{z}_{t+1}\|^2].$$

It then follows the convergence theorem for Algorithm 1.

**Theorem 2.7.** *Under Assumption 2.2, 2.3 and with a proper settings of parameters $\eta_1, \eta_2, \beta_1 = \mathcal{O}(B\epsilon^2)$, $\beta_0 = \mathcal{O}(\min\{|I_t|, B\}\epsilon^2)$ and $\eta_0 = \mathcal{O}\left(\min\left\{\min\{|I_t|, B\}\epsilon^2, \frac{B|I_t|\epsilon^2}{m}\right\}\right)$, Algorithm 1 ensures that after $T = \mathcal{O}\left(\max\left\{\frac{m}{|I_t|B\epsilon^4}, \frac{1}{\min\{|I_t|, B\}\epsilon^4}\right\}\right)$ iterations we can find an $\epsilon$-stationary solution of $F(\mathbf{x})$, i.e., $\mathbb{E}[\|\nabla F(\mathbf{x}_\tau)\|^2] \leq \epsilon^2$ for a randomly selected $\tau \in \{0, \ldots, T\}$.*

### 2.2.2 Convergence analysis of Algorithm 2

Similarly, to bound the gap between the true gradient $\nabla F(\mathbf{x}_t)$ and its estimator $\mathbf{z}^{t+1}$ in Algorithm 2, we aim to bound the following three errors, $\delta_{y,t}$, $\delta_{\alpha,t}$ and $\sum_{i \in \mathcal{S}} \|\mathbf{v}_i^t - [\nabla_{yy}^2 g_i(\mathbf{x}_t, \mathbf{y}_i(\mathbf{x}_t))]^{-1} \nabla_y f_i(\mathbf{x}_t, \alpha_i(\mathbf{x}_t), \mathbf{y}_i(\mathbf{x}_t))\|^2 =: \delta_{\mathbf{v},t}$. We follow the same strategy for $\delta_{y,t}$ and $\delta_{\alpha,t}$ to what has been discussed in the previous section. To deal with $\delta_{\mathbf{v},t}$, we first note that by its construction, $\gamma_i(v, \mathbf{x}, \mathbf{y}_i)$ is $\mu_g$-strongly convex and $L_g$-smooth with respect to $v$. At a result, similarly to Lemma 2.5, one may bound the error of the estimators $\mathbf{v}_i^t$. Then it follows the convergence theorem for Algorithm 2.

**Theorem 2.8.** *Considering Algorithm 2, under Assumption 2.2, 2.3 and with a proper settings of parameters* $\eta_1, \eta_2, \eta_3 = \mathcal{O}\left(B\epsilon^2\right)$, $\beta_0 = \mathcal{O}\left(\min\{|I_t|, B\}\epsilon^2\right)$ *and* $\eta_0 = \mathcal{O}\left(\min\left\{\min\{|I_t|, B\}\epsilon^2, \frac{B|I_t|\epsilon^2}{m}\right\}\right)$, *Algorithm 1 ensures that after* $T = \mathcal{O}\left(\max\left\{\frac{m}{|I_t|B\epsilon^4}, \frac{1}{\min\{|I_t|, B\}\epsilon^4}\right\}\right)$ *iterations we can find an $\epsilon$-stationary solution of $F(\boldsymbol{x})$, i.e., $\mathbb{E}[\|\nabla F(\boldsymbol{x}_\tau)\|^2] \leq \epsilon^2$ for a randomly selected $\tau \in \{0, \dots, T\}$.*

**Remark.** In Theorem 2.7 and Theorem 2.8, there is no condition on the sizes of data batches $\mathcal{B}_i^t, \tilde{\mathcal{B}}_i^t$ nor block batch $I_t$ for the algorithm to converge. Hence, their sizes can be as small as one. The order of complexity is $\mathcal{O}(1/\epsilon^4)$, which matches the optimal complexity for nonconvex optimization under a general unbiased stochastic oracle model. In addition, there is parallel speed up by increasing batch sizes for data samples and task samples due to the scaling in terms of $|I_t|$ and $B$ in the iteration complexity.

## 3 Applications in Multi-task Deep (Partial) AUC Maximization

In this section, we present two applications of multi-block min-max bilevel optimization: multi-task deep AUC maximization and deep partial AUC maximization. (partial) AUC is a performance measure of classifiers for imbalanced data. Recent studies have shown great success of deep AUC maximization in various domains (e.g., medical image classification and molecular property prediction) [26, 33, 40]. However, efficient algorithms for multi-task deep (partial) AUC maximization have not been well developed. For multi-task deep AUC maximization, we solve an existing formulation by our algorithm. For multi-task deep partial AUC maximization, we propose a new bilevel formulation and solve it by our algorithm.

### 3.1 Muti-task Deep AUC maximization

Following the previous work [26, 40], deep AUC maximization problem can be formulated as a non-convex strongly concave min-max optimization problem $\min_{\mathbf{w},a,b} \max_{\alpha \in \mathcal{A}} L_{\text{AUC}}(\mathbf{w}, a, b, \alpha)$. However, training a deep neural network from scratch by optimizing AUC loss does not necessarily lead to a good performance[39]. To address this issue, [39] proposed a compositional training strategy for deep AUC maximization:

$$\min_{\mathbf{w},a,b} \max_{\alpha \in \mathbb{R}_+} L_{\text{AUC}}(\mathbf{w} - \tilde{\eta} \nabla L_{\text{CE}}(\mathbf{w}), a, b, \alpha),$$

where $L_{\text{CE}}$ denotes the cross-entropy loss. The outer objective remains to be the AUC loss, while the inner objective is a gradient descent step of minimizing the traditional cross-entropy loss. This method has shown superior performance on various datasets [39]. We extend this formulation to multi-task problems and reformulate it into a multi-block min-max bilevel optimization:

$$\min_{(\mathbf{w}_l, \mathbf{w}_h), \mathbf{a}, \mathbf{b}} \max_{\boldsymbol{\alpha} \in \mathbb{R}_+^m} \sum_{i=1}^m L_{\text{AUC}}(\mathbf{u}^i(\mathbf{w}_l, \mathbf{w}_h), a^i, b^i, \alpha^i)$$

$$\text{s.t. } \mathbf{u}^i(\mathbf{w}_l, \mathbf{w}_h) = \arg\min_{\mathbf{u}^i} \frac{1}{2} \left\| \mathbf{u}^i - ((\mathbf{w}_l, \mathbf{w}_h^i) - \tilde{\eta} \nabla L_{\text{CE}}(\mathbf{w}_l, \mathbf{w}_h^i)) \right\|^2,$$

where $\mathbf{w}_l$ denotes the weight for the encoder network that is shared for all tasks, and $\mathbf{w}_h = (\mathbf{w}_h^1, \dots, \mathbf{w}_h^m)$ denote the task-owned classification heads. The upper objective is strongly concave in terms of dual variables $\alpha_i$ and the lower level objective is strongly convex in terms of $\mathbf{u}^i$. The hessian of the lower-level objective is the identity matrix. Hence, there is no need to track and estimate the hessian matrix.

## 3.2 Multi-task Deep Partial AUC Maximization

Some real-world applications (e.g., medical diagnosis [2]) cannot tolerate a model with a high False Positive Rate (FPR) even though it has significant performance in AUC. Hence, a measure of interest is one-way partial AUC (pAUC), which puts a restriction on the range of FPR (i.e., FPR$\in [\rho_l, \rho]$, where $0 \leq \rho_l \leq \rho \leq 1$). Below, we focus on the case $\rho_l = 0$. However, our method can be easily extended for handling $\rho_l > 0$. Let $\mathcal{D}_+, \mathcal{D}_-$ denote the set of positive and negative data for a binary classification task, respectively. Let $\mathcal{D}_-[K]$ denote the top-K negative examples according to their prediction scores. Let $n_+, n_-$ denote the number of positive and negative samples respectively. Then we have partial AUC optimization with a pairwise square loss formulated as following [37]:

$$\min_{\mathbf{w}} \frac{1}{n_+} \sum_{\mathbf{x}_i \in \mathcal{D}_+} \frac{1}{n_- \rho} \sum_{\mathbf{x}_j \in \mathcal{D}_-[K]} (h_{\mathbf{w}}(\mathbf{x}_j) - h_{\mathbf{w}}(\mathbf{x}_i) + c)^2,$$

where $K = n_- \rho$, $c$ is a constant and $h_{\mathbf{w}}(\cdot)$ denotes the prediction score on a data. A key challenge for solving the above problem is to deal with the non-differentiable top-K selector $\mathbf{x}_j \in \mathcal{D}_-[K]$, which depends on the model parameters $\mathbf{w}$. This challenge has been recently tackled in [38, 42]. We focus on the comparison with the first work as it is optimization oriented similar to ours and also has the state-of-the-art performance. They formulate the problem into either a weakly convex minimization or approximate it by a smooth objective in a compositional form. A caveat of their algorithms (named SOPA, SOPA-s) is that they need to maintain and update $n_+$ auxiliary variables with one for each positive data. If we apply their algorithms for multi-task problems, one needs to maintain and update $\sum_{i=1}^m n_+^i$ auxiliary variables, which could dramatically slow down the convergence.

To address this problem, we first transform it into a min-max optimization problem. Let $a(\mathbf{w}) = \frac{1}{n_+} \sum_{\mathbf{x}_i \in \mathcal{D}_+} h_{\mathbf{w}}(\mathbf{x}_i)$ and $b(\mathbf{w}) = \frac{1}{n_- \rho} \sum_{\mathbf{x}_j \in \mathcal{D}_-[K]} h_{\mathbf{w}}(\mathbf{x}_j)$. Then we can write the problem as (cf. Appendix B for a derivation)

$$\min_{\mathbf{w}, a, b} \frac{1}{n_+} \sum_{\mathbf{x}_i \in \mathcal{D}_+} (h_{\mathbf{w}}(\mathbf{x}_i) - a)^2 + \frac{1}{n_- \rho} \sum_{\mathbf{x}_j \in \mathcal{D}_-} \mathbb{I}(\mathbf{x}_j \in \mathcal{D}_-[K])(h_{\mathbf{w}}(\mathbf{x}_j) - b)^2 + (b(\mathbf{w}) - a(\mathbf{w}) + c)^2.$$

To tackle non-continuous non-differentiable indicator function $\mathbb{I}(\cdot)$ we can replace it by a sigmoid function. To tackle non-differentiability of the top-K selector $\mathbf{x}_j \in \mathcal{D}_-[K]$, we follow [29] and formulate it as lower-level optimization problem, i.e., $\mathbf{x}_j \in \mathcal{D}_-[K]$ is equivalent to $h_{\mathbf{w}}(\mathbf{x}_j) > \lambda(\mathbf{w})$, where $\lambda(\mathbf{w})$ represents the $K + 1$-th largest scores among all negative examples, which can be approximated by a solution from a smooth strongly convex minimization problem as following:

$$\lambda(\mathbf{w}) = \arg\min_{\lambda \in \mathbb{R}} L(\lambda, \mathbf{w}) := \frac{K + \varepsilon}{n_-} \lambda + \frac{\tau_2}{2} \lambda^2 + \frac{1}{n_-} \sum_{\mathbf{x} \in \mathcal{D}_-} \tau_1 \ln(1 + \exp((h_{\mathbf{w}}(\mathbf{x}) - \lambda)/\tau_1)),$$

where $\varepsilon, \tau_1, \tau_2$ are small constants. Based on above, the multi-task deep partial AUC minimization problem can be formulated as a multi-block min-max bilevel optimization problem give by:

$$\min_{\mathbf{w}, \mathbf{a}, \mathbf{b}} \max_{\boldsymbol{\alpha} \in \mathbb{R}^m} \sum_{k=1}^m \left\{ \frac{1}{n_+^k} \sum_{\mathbf{x}_i \in \mathcal{D}_+^k} (h_{\mathbf{w}}(\mathbf{x}_i; k) - a_k)^2 + \frac{1}{n_-^k \rho} \sum_{\mathbf{x}_j \in \mathcal{D}_-^k} \phi(h_{\mathbf{w}}(\mathbf{x}_j; k) - \lambda_k(\mathbf{w}))(h_{\mathbf{w}}(\mathbf{x}_j; k) - b_k)^2 \right.$$

$$\left. + 2\alpha_k \left( \frac{1}{n_-^k \rho} \sum_{\mathbf{x}_j \in \mathcal{D}_-^k} \phi(h_{\mathbf{w}}(\mathbf{x}_j; k) - \lambda_k(\mathbf{w})) h_{\mathbf{w}}(\mathbf{x}_j; k) - \frac{1}{n_+^k} \sum_{\mathbf{x}_i \in \mathcal{D}_+^k} h_{\mathbf{w}}(\mathbf{x}_i; k) + c \right) - \alpha_k^2 \right\}$$

$$\lambda_k(\mathbf{w}) = \arg\min_{\lambda \in \mathbb{R}} L_k(\lambda, \mathbf{w}) := \frac{K + \varepsilon}{n_-} \lambda + \frac{\tau_2}{2} \lambda^2 + \frac{1}{n_-} \sum_{\mathbf{x}_j \in \mathcal{D}_-^k} \tau_1 \ln(1 + \exp((h_{\mathbf{w}}(\mathbf{x}_j; k) - \lambda)/\tau_1)),$$

where $\mathcal{D}_{+/-}^k$ denote the positive/negative data set of the $k$-th task, $h_{\mathbf{w}}(\mathbf{x}; k)$ denote the prediction score for the $k$-th classifier, and $\phi(s) = \frac{1}{1 + \exp(-s)}$ is the sigmoid function. The upper-level objective is strongly concave in terms of $\alpha_k$ and the lower-level objective is strongly convex in terms of $\lambda_k$.

We develop a tailored algorithm based on Algorithm 1 for solving the above formulation of multi-task pAUC maximization as shown in Algorithm 3 in Appendix C. For the hessian update, the momentum

Table 1: The testing AUC scores on four datasets.

| Method\DataSet | CIFAR100 | CheXpert | CelebA | ogbg-molpcba |
|---|---|---|---|---|
| mAUC (baseline) | 0.9044 (0.0015) | 0.8084( 0.1455) | 0.9062 (0.0042) | 0.7793(0.0028) |
| mAUC-CT (ours) | **0.9272 (0.0014)** | **0.8198(0.1495)** | **0.9192 (0.0004)** | **0.8406(0.0044)** |

update (2) is efficient due to that the each lower-level problem is only one-dimensional. For simplicity of implementation, we define a loss $G^{t+1}$ 38 in the Appendix C, on which auto-differentiation can be directly applied for computing a gradient estimator. We refer readers to Appendix C for a detailed explanation and derivation of $G^{t+1}$.

# 4 Experiments

## 4.1 Multi-task Deep AUC Maximization with Compositional Training

**Data.** We use four datasets, namely CIFAR100, CheXpert, CelebA and ogbg-molpcba. CIFAR-100 [22] is an image dataset consisting of $60,000$ $32 \times 32$ color images in 100 classes. Hence, there are 100 tasks for CIFAR100. We follow $45,000/5,000/10,000$ split to construct training/validation/testing datasets. CelebA [27] is a large-scale face attributes dataset with more than 200K celebrity images, each with 40 attribute annotations (i.e., 40 tasks). We use the recommended training/validation/testing split as $162,770/19,866/19,961$. CheXpert [18] is a dataset that contains 224,316 chest radiographs with 14 observations. Since the official testing dataset is not open to public, we take the official validation set as the testing data, and take the last 1000 images in the training dataset for validation. Due to the absence of positive samples for the observation *Fracture* in the testing dataset, we ignore this label and only consider the rest 13 observations (i.e., 13 tasks). The last dataset ogbg-molpcba is a molecular property prediction graph dataset [15]. It consists of 437,929 graphs with 128 labels (i.e., 128 tasks). We follow scaffold splitting procedure as recommended in [34].

**Models.** We use ResNet18 [13] for CIFAR-100 and CelebA, and ImageNet pretrained DenseNet121 [17] for CheXpert. For ogbg-molpcba, we use Graph Isomorphism Network (GIN) [35].

**Setup.** We compare our method for optimizing the multi-task AUC maximization with compositional training denoted by mAUC-CT (ours) with a baseline that directly optimizes multi-task min-max AUC loss denoted by mAUC (baseline). We do not compare with other straightforward baselines (e.g., optimizing the CE loss and the focal loss) since they have been shown to be inferior than AUC maximization methods for imbalanced data in many previous works [40, 42]. For both methods, the learning rates $\eta_2, \eta_1, \eta_0$ are set to be the same and tuned in $\{0.01, 0.03, 0.05, 0.07, 0.1\}$. The learning rates decay by a factor of 10 at the 4th and 30th epoch for CheXpert and CelebA, respectively. No learning rate decay is applied for CIFAR-100 and ogbg-molpcba. The moving average parameter $\beta_0$ and $\tilde{\eta}$ in the lower level problem of mAUC-CT (ours) are tuned in $\{0.1, 0.5, 0.9\}$. Regarding the task sampling, for datasets CIFAR-100 and ogbg-molpcba, 10 tasks are sampled to be updated in each iteration, and for each sampled task, we independently sample a data batch of size 128. For the other two datasets with fewer tasks, CheXpert and CelebA, we sample one task at each iteration. The batch size for data samples is 32 for CheXpert, and 128 for CelebA. We run both methods the same number of epochs which varies on different data, 2000 epochs for CIFAR100, 6 epochs for CheXpert, 40 epochs for CelebA and 100 epochs for obgb-molpcba.

**Results.** In Table 1, we report the testing AUC score with the model selected according to the best performance on validation datasets. Comparing with optimizing AUC loss directly, mAUC-CT (ours) achieves better performance on all tested datasets. We show the results of an ablation study in Figure 3, which verifies convergence has a parallel speed-up effect on both the batch sizes of data samples and task samples. The algorithm converges faster as either the data or task sample batch size increases. In Figure 2 (left two), we compare our method with the baseline in terms of convergence speed on the training data of two datasets, which demonstrate that our method converges faster. More results on training convergence are included in the appendix.

## 4.2 Multi-task Deep Partial AUC Maximization

**Setup.** For this task, we use the same datasets and the same networks for the image datasets and graph datasets as in the previous subsection. For baselines, we compare with a naive mini-batch based method (MB) for pAUC maximization [20], and a state-of-the-art pAUC maximization method SOPA-s [42]. Following previous works [40, 42], for pAUC maximization methods we use a pretrained

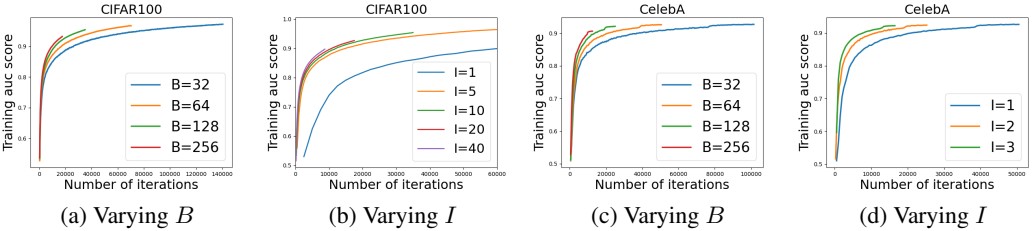

(a) Varying $B$       (b) Varying $I$       (c) Varying $B$       (d) Varying $I$

Figure 1: Convergence of our method vs data sample batch sizes $B$ and vs task sample batch size $I := |I_t|$ for multi-task deep AUC maximization.

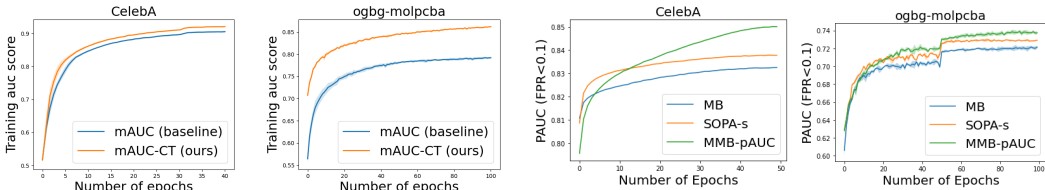

Figure 2: Comparison of Convergence on training data for multi-task deep AUC maximization (left two) and multi-task deep pAUC maximization (right two) on the CelebA and ogbg-molpcba datasets.

encoder network by optimizing the CE loss as the initial encoder and learn the whole network by maximizing pAUC. We also report the performance of optimizing the CE loss for a refernece. For all methods, the learning rate is tuned in $\{0.0001, 0.0005, 0.001, 0.005, 0.01\}$. The hyperparameters selection of MMB-pAUC are: $\eta_1$ and $\eta_2 \in \{0.5, 0.1, 0.01\}$, $\beta_1 \in \{0.99, 0.9, 0.5, 0.1, 0.01\}$ and $\beta_0 \in \{0.9, 0.99\}$. For Focal loss we select gamma from $\{1, 2, 4\}$ and alpha from $\{0.25, 0.5, 0.75\}$. The momentum parameters in SOPA-s are tuned in the same range and their $\lambda$ parameter in $\{0.1, 1, 10\}$ as in [42]. The margin parameter in the surrogate loss (e.g., $c$) is set to be 1. Regarding the task sampling, we sample one task at each iteration for ogbg-molpcba and CheXpert, sample 10 tasks for CIFAR100, and sample 4 tasks for CelebA. The data sample batch size is 32 for CheXpert, and 64 for others. For smaller datasets (CIFAR100 and ogbg-molpcba), we run 100 epochs for each, and we decay the learning rate by a factor of 10 at the 50-th epoch. For larger datasets (CelebA, CheXpert), we run 50 and 5 epochs respectively.

**Results.** The partial AUC scores with FPR$\leq 0.1$ on the testing data of different methods are shown in Table 2. From the results, we can see that our methods perform better than baseline methods with a significant margin. In Figure 2 (right two), we compare our method with the baselines in terms of convergence speed on the training data of two datasets, which demonstrate that our method converges faster. More results on training convergence are included in the appendix.

Table 2: The testing partial AUC scores on the four datasets.

| Method\DataSet | CIFAR100 | CelebA | CheXpert | ogbg-molpcba |
|---|---|---|---|---|
| CE | 0.8895 (0.0009) | 0.8024 (0.0026) | 0.6606 (0.0159) | 0.6576 (0.0010) |
| Focal | 0.8966 (0.0007) | 0.8064 (0.0011) | 0.6646 (0.0132) | 0.6453 (0.0021) |
| MB | 0.9188 (0.0006) | 0.8304 (0.0005) | 0.6759 (0.0160) | 0.7213 (0.0018) |
| SOPA-s | 0.9251 (0.0003) | 0.8336 (0.0001) | 0.6682 (0.0156) | 0.7290 (0.0019) |
| Ours | **0.9262 (0.0005)** | **0.8360 (0.0003)** | **0.6827 (0.0183)** | **0.7374 (0.0015)** |

## 5 Conclusion and Future Work

We have developed two simple single loop randomized stochastic algorithms for solving multi-block min-max bilevel optimization problems. These algorithms require updates for only constant number of blocks in each iteration. We showed that both of them achieve an oracle complexity of $\mathcal{O}(\epsilon^{-4})$, which matches the optimal complexity order for solving stochastic nonconvex optimization under a general unbiased stochastic oracle model. At the same time, we hope our work inspires others to find more novel applications of our idea.

## Acknowledgments and Disclosure of Funding

This work is partially supported by NSF awards 2147253, 2110545, 1844403, and Amazon research award.

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
