# OpenReview forum: "Multi-block Min-max Bilevel Optimization with Applications in Multi-task Deep AUC Maximization"
_NeurIPS.cc/2022/Conference — NeurIPS 2022 Accept_

### Official Review · Reviewer_Z9r5 · 2022-07-09

**Rating:** 8
**Confidence:** 5
**Soundness:** 4 excellent
**Presentation:** 3 good
**Contribution:** 3 good

**Summary:**

This paper investigated how to optimize multi-task deep auc maximization problems. In particular, this paper proposed a novel efficient algorithm for multi-block min-max bilevel optimization problems. The theoretical analysis demonstrated that this algorithm can achieve $O(1/\epsilon^4)$ convergence rate, matching the optimal complexity for nonconvex optimization. The empirical results confirm the efficacy of the proposed algorithm.

Overall, this algorithm is novel and it is well reasoned with practical applications. The theoretical analysis is also solid.  Thus, this is a strong submission with significant theoretical and empirical contributions.

**Questions:**

- In the experiment, is a class viewed as a task?
- It would be good if Section 3 could be moved before Section 2.1. Then, it will be much clear what applications belong to this kind of problem.
- It looks like there is a typo in Line 526: Assumption ??


====post rebuttal=====

Thanks for your answers.

This is a novel method for the important AUC maximization problem. It can benefit a lot of real-world applications. Thus, I am raising my rating and suggest acceptance.

**Strengths And Weaknesses:**

Strengths:
- The proposed algorithm is novel. It provides an efficient solution for optimizing the important AUC maximization problem. It should have wide applications in real-world data analysis tasks because the dataset in real-world applications is typically imbalanced.
- The technical contributions are solid. The authors provide a rigorous analysis for the convergence rate. And the convergence rate can match the optimal complexity of standard nonconvex optimization problems.
- The extensive experiments with different datasets and models have shown better performance than existing methods. This confirms the efficacy of the proposed algorithm.


Weaknesses:
- In the experiment, is a class viewed as a task?
- It would be good if Section 3 could be moved before Section 2.1. Then, it will be much clear what applications belong to this kind of problem.
- It looks like there is a typo in Line 526: Assumption ??

---

> ### Author Response · Authors · 2022-07-29
> **Regarding Reviewer Z9r5’s Concerns**
>
> Thank you for your positive rating and constructive comments! We have revised the paper accordingly.  Below we address your questions.
>
> **Q1**: In the experiment, is a class viewed as a task?\
> **Response**: Yes, each class is a task.
>
>
> **Q2**: It would be good if Section 3 could be moved before Section 2.1. Then, it will be much clear what applications belong to this kind of problem.\
> **Response**: Thank you for the suggestion! We understand that introducing applications earlier will make the motivation of proposing new method clearer. However, since Section 3 contains not only the application description, but also Algorithm 2 (MMB-pAUC), placing this section before Section 2.1 seems to be inappropriate. Thus we decide to clarify the contents for each section in the contribution part (line 83 - line 94). We hope this would be helpful directing the readers.

---

### Official Review · Reviewer_hKcV · 2022-07-11

**Rating:** 7
**Confidence:** 3
**Soundness:** 3 good
**Presentation:** 2 fair
**Contribution:** 3 good

**Summary:**

This paper studies a bilevel multi-block (multi-objective), where the inner level is a per-block convex minimization problem, while the outer level is a min-max problem involving multiple blocks. The paper proposes a novel algorithm which contains existing algorithms for min-max optimization and single-block bilevel optimization as special cases. This algorithm makes use of non-standard technique to obtain the implicit gradient for the outer level variable, and enhances the outer level optimization with a momentum based updated. This algorithm is then analyzed in terms of the convergence to a stationary point, and key aspects of this novel analysis is highlighted and discussed. The main convergence rate matches the optimal convergence rate for stochastic nonconvex min-max and bilevel optimization separately, highlighting the tightness of the result. The paper then proposes two related applications of this multi-block bilevel problem -- multi-task (partial) AUC maximization for training neural networks. One non-trivial contribution in the application of the multi-block bilevel problem to partial AUC maximization is a novel formulation of partial AUC maximization as a min-max bilevel problem. Empirical evaluation of the proposed algorithm on these applications with data from different domains (images, graphs) and different neural network architectures highlight the improved performance of the proposed scheme against various baselines.


**Questions:**

- Regarding [8] considering "a special case where $f(\mathbf{x}, \cdot, \mathbf{y})$ is a linear function", this is infact true. But there is an assumption (Assumption 2.2)  on strong concavity w.r.t. $\alpha$, and I think the linear form of [8] does not satisfy that assumption. So does this mean that the $\min_{\mathbf{x}} \max_{i \in [1, \ldots, m]} f_i(\mathbf{x}, y_i(x))$ $\equiv$ $\min_{\mathbf{x}} \max_{\alpha \in \Delta_m \subset \mathbb{R}^m} \alpha_i \cdot f_i(\mathbf{x}, y_i(x))$ does not fit in the version of problem (1) studied in this paper? Note that $\Delta_m$ is the $m$-dimensional simplex implying $\alpha_i \geq 0 \forall i \in [1, \ldots m]$ and $\sum_{i = 1}^m \alpha_i = 1$.

- The $\max_\alpha$ in AUC maximization is there even for the single block problem so it is not clear what is the effect of this $\max_{\boldsymbol{\alpha}}$ in the multi-task setting. It does not appear that the $\boldsymbol{\alpha}$ is forcing some form of robustness across the tasks as something like the $\min_{\mathbf{x}} max_i$ of [8] would have. Is that a correct assessment? Can you please elaborate of the effect (if any) of the $\max_{\boldsymbol{\alpha}}$ in the multi-block setting?

- Is the main contribution in subsection 4.3 the formulation of multi-task pAUC maximization as problem (1)?


**Limitations:**

The authors do explicitly discuss one limitation of their work. I do not think there are any potential negative societal impact of this work.


**Strengths And Weaknesses:**

### Strengths:

- This paper studies a novel generalization of the (recently well-studied in machine learning literature) bilevel problem to multiple objectives. The problem (1) is infact a more general form of recent problem studied in [8]. Beyond the more general problem, the proposed algorithm also makes use of a momentum enhanced gradient update that is yet another novelty over the recently proposed [8] for the min-max multi-block bilevel problem.


- The authors do a great job at positioning the problem and the novel results against existing literature on bilevel, min-max, multi-block bilevel problem. This level of clarity makes it easy to understand the originality in the paper.

- The discussion of the main theoretical results clearly highlights the various strengths of the analysis. One such strength is that the convergence does not require a minimal data batch or block-batch size. Furthermore, the result shows that the algorithm can benefit from increasing both the data batch and task batch size. This is very nice property of the algorithm and analysis, and this property was empirically demonstrated with experiments.


- The empirical evaluation considers various datasets from different domains and different neural network architectures, highlighting the generality of the proposed problem and algorithm.


### Weaknesses:

- It is not clear with no common application such as those in [8] of representation learning or hyperparameter optimization (which are multi-task extensions of the well-studied bilevel problems) are considered in this paper to evaluate the proposed general algorithm.

- One aspect of the proposed algorithm that estimates the outer level gradient with equations (2) and (3) is that it is not clear how feasible this scheme is for practical problems without a special structure (like the hessian $\nabla_{yy}^2 g_i$  is the Identity matrix). The Neumann series allows one to approximate the matrix inversion with a series of more practically efficient matrix-vector product. But the proposed scheme requires one to take the full inverse for each iteration. The authors mention this weakness as one of the limitations.

- Various aspects of the formulation of the partial AUC maximization as problem (1) is unclear. First, in the vanilla AUC maximization problem, there is a positive $\alpha$ even for the single-task (or single-block) problem. However, in the reformulation of the partial AUC maximization problem in the equations between lines 253-254 and 258-259, there is no $\alpha$ present in the single-task problem. Then, in the multi-task extension in equation after line 260 on page 7, $\boldsymbol{\alpha}$ and $\alpha_k, k \in [1, \ldots, m]$ are introduced without any context, and the problem itself appears to have various typographical errors since the $\max_{\boldsymbol{\alpha} \in \mathbb{R}^m}$ appears to be outside the $\sum_{k = 1}^m$. Moreover, it is not clear why we dont need  $\max_{\boldsymbol{\alpha} \in \mathbb{R}_+^m}$ (implying positive $\alpha_k$s).


- Given the decoupled nature of the $\alpha_k$s in the (partial) AUC maximimization problem, and the assumptions on problem (1), it is not clear why the problem cannot be formulated a standard min-max bilevel optimization problem:


$$
\min_{\mathbf{x}} \max_{\boldsymbol{\alpha} \in \mathcal{A}} F(\mathbf{x}, \boldsymbol{\alpha}, \mathbf{y}(\mathbf{x})) := \sum_{i = 1}^m f_i(\mathbf{x}, \alpha_i, \mathbf{y}_i(\mathbf{x})
$$
with

$$
\mathbf{y}(\mathbf{x}) = \arg \min_{\mathbf{y}:= \lbrace y_i , i \in [1, \ldots, m] \rbrace } \sum_{i = 1}^m g_i(\mathbf{x}, y_i).
$$

One difference is in the proposed algorithm is that only a subset of the blocks are chosen, but in the extreme where all the blocks are chosen in each iteration, this is just a standard min-max bilevel solver such as the one that might be used to solve the single-task version of the AUC maximization problem listed in the equation between lines 223-224. If this assessment is true, I think it would be good to understand what improvement the proposed problem formulation and algorithm brings over just looking at the multi-block problem as a single min-max bilevel optimization problem and solving it thusly.

---

> ### Author Response · Authors · 2022-07-29
> **Regarding Reviewer hKcV’s Concerns**
>
> Thank you for your positive rating! Below we address your questions.
>
> **Q1**: Does the formulation in [8] fits in the general formulation considered in this work?\
> **Response**: No. Since our formulation requires $f_i(x,\cdot,y_i)$ to be strongly concave, the linear formulation in [8] does not fit in our framework.
>
>
> **Q2**: It does not appear that the $\alpha$ is forcing some form of robustness across the tasks as something like the $\min_x \max_i$ of [8] would have. Is that a correct assessment? Can you please elaborate of the effect (if any) of the $\max_\alpha$ in the multi-block setting?\
> **Response**: You are correct, here in AUC maximization the $\max_\alpha$ has nothing to do with robustness. Unlike the multi-task representation learning model from [8], the $\max_\alpha$ in multi-task AUC maximization is part of the AUC maximization, which for one task is formulated as a min-max problem:
>
>
> $$ \min_{\textbf{w},a,b}\max_{\alpha\in \mathbb{R}_+} L (\textbf{w},a,b,\alpha)\quad \quad \text{(single-task case)}$$
>
> where $L$ is the AUC surrogate loss. The effect of the multi-block setting  is that there are  multiple blocks in $\alpha$, i.e., $\alpha=(\alpha_1, \ldots, \alpha_m)$ and the update of all blocks will depend on data for all tasks. Hence, for a large number of tasks it is not memory and computationally efficient to sample and process data for all tasks and  it is necessary to develop efficient algorithms to update a few blocks of $\alpha$ at each iteration by sampling data for the sampled tasks only.
>
>
> **Q3**: Is the main contribution in subsection 4.3 the formulation of multi-task pAUC maximization as problem (1)?\
> **Response**: Yes.
>
>
> **Q4**: It is not clear why no common application such as those in [8] of representation learning or hyperparameter optimization (which are multi-task extensions of the well-studied bilevel problems) are considered in this paper to evaluate the proposed general algorithm.\
> **Response**: Since our formulation requires the objective of $\max_\alpha$ to be strongly concave, applications such as those considered in [8] do not fit in.
>
>
> **Q5**: The proposed scheme requires one to take the full inverse for each iteration.\
> **Response**: This is indeed a limitation that we concerned about. However, for the considered applications this is not a problem as the Hessian is either an identity matrix (in sec. 3.1) or a scalar (in sec 3.2). We have managed to tackle this problem by a minor change to the algorithm. The theoretical convergence rate remains to be $1/\epsilon^4$ while computing matrix inverse is no longer needed. We add this algorithm and its convergence analysis in the Appendix D for your reference. The main idea is to treat $[\nabla_{yy}^2 g_i(\textbf{x},\textbf{y}_i)]^{-1} \nabla_y f_i(\textbf{x},\alpha_i,\textbf{y}_i)$
>
> as the solution to a quadratic function minimization problem. As a result, $[\nabla^2_{yy}g_i(\textbf{x},\textbf{y}_i)]^{-1}\nabla_y f_i(\textbf{x},\alpha_i,\textbf{y}_i)$ can be approximated by SGD. In other words, it is treated as another lower-level problem.
>
>
> **Q6**: About multi-task pAUC formulation. In the multi-task extension in equation after line 260 on page 7,  $\alpha$ and $\alpha_k$ are introduced without any context.\
> **Response**: We apologize for the confusion on this part. We omit some details due to the limited space. The $\alpha$ terms in line 260 are derived from the last term $(b(\textbf{w})-a(\textbf{w})+c)^2$ in line 253.\
> Due to the fact that $(b(\mathbf w)-a(\mathbf w)+c)^2$ cannot be directly obtained since $a(\mathbf w)$ and $b(\mathbf w)$ are expectations, one may use $p^2 = \max_\alpha 2p\alpha-\alpha^2$ to get
>
> $$(b(\mathbf w)-a(\mathbf w)+c)^2=\max_\alpha \mathbb{E}_{\textbf{x}_j\in \mathcal{D}-[K],\,\textbf{x}_i\in \mathcal D+}[2\alpha(h_\textbf{w}(\textbf{x}_j)-h_\textbf{w}(\textbf{x}_i)+c)-\alpha^2]$$
>
> Then by extending the formulation to the multi-task setting and replacing the top-K selector with $\phi(h_{\textbf{w}}(\textbf{x}_j;k)-\lambda_k(\textbf{w}))$, we obtain the formulation in line 260. We have added more details in the appendix (lines 565 - lines 569).
>
>
> **Q7**: $\max_{\alpha\in\mathbb{R}^m}$ should be outside the $\sum_{k=1}^m$.\
> **Response**: Thank you for pointing out this mistake. We  have fixed it in the revision.
>
>
> **Q8**: It is not clear why we dont need $\max_{\alpha\in\mathbb{R}^m_+}$
>  (implying positive $\alpha_k$s).\
> **Response**: Both  $\alpha\in\mathbb R^m_+$  and $\alpha\in\mathbb R^m$  has been used for AUC maximization. Please refer to [38]. For deep partial AUC maximization, since we directly derive the equivalent min-max formulation from the square loss under line 241 hence $\alpha\in\mathbb R^m$ is the result.

---

> > ### Comment · Reviewer_hKcV · 2022-08-07
> > **Response received, minor follow-up**
> >
> > Thank you for the detailed response. I also appreciated the **Novelty for handling the maximization problems for deriving $O(1/\epsilon^4)$ complexity** response to reviewer **h3Nq**.
> >
> > Given the problem studied here, based on your responses to **Q1**, **Q2** and **Q4**,  what are some other applications other than (multi-task) AUC/pAUC maximization where such a multi-block (for $\alpha$) single model bilevel problem would appear?

---

> > > ### Author Response · Authors · 2022-08-08
> > > **Thank you!**
> > >
> > > Thank you for acknowledging our responses.
> > >
> > > Below we give another example of multi-block min-max bilevel problems.  Let us consider multi-task bipartite ranking problems with the p-norm push objective (Rudin (2009). The P-Norm Push: A Simple Convex Ranking Algorithm that Concentrates at the Top of the List). The p-norm push objective is given by $\sum_{\mathbf x_j\in\mathcal D_-}g(\sum_{\mathbf x_i\in\mathcal D_+}\ell(h_{\mathbf w}(\mathbf x_i) - h_{\mathbf w}(\mathbf x_j)))$, where $h_{\mathbf w}(\mathbf x)$ denotes the prediction score of a network on a data,  $\mathcal D_+$ ($\mathcal D_-$) denotes the set of positive (negative) examples, and $g(\cdot) = (\cdot)^p$ for some $p>1$, which is convex for non-negative inputs. As a result, we can use the convex conjugate of $g$ to transform the problem into a min-max formulation, i.e., $\sum_{\mathbf x_j\in\mathcal D_-}\max_{\alpha}\alpha \sum_{\mathbf x_i\in\mathcal D_+}\ell(h_{\mathbf w}(\mathbf x_i) - h_{\mathbf w}(\mathbf x_j)) - g^*(\alpha)$. Then by combining the above loss for multiple tasks, we get a min-max formulation with multiple blocks of $\alpha$, where each block of $\alpha$ depends on a particular task. The bilevel formulation will arise when we want to learn a good feature representation using the compositional training method proposed in Yuan et al. 2021 [37], which can be formulated as a bilevel formulation  similar to what we considered for multi-task deep AUC maximization in the paper.
> > >
> > > We hope this example makes sense to you. Thank you!

---

> > > > ### Comment · Reviewer_hKcV · 2022-08-09
> > > > **Thank you for the example**
> > > >
> > > > Thank you for the new example. If I understand correctly both these applications (the one here and the ones in the paper) are where the $\alpha$ term appears as the dual variable for some objective, and the "bilevel" nature is introduced via separating the features and the models, and the multi-task/block appears via sharing the representations and having a task/block specific model in the shared representation space. Is that an accurate (albeit oversimplified) summary?

---

> > > > > ### Author Response · Authors · 2022-08-09
> > > > > **You are correct!**
> > > > >
> > > > > Thank you!

---

> ### Author Response · Authors · 2022-07-29
> **Regarding Reviewer hKcV’s Concerns**
>
> **Q9**:  Is the considered problem a standard min-max bilevel optimization problem with a single block? It would be good to understand what improvement the proposed problem formulation and algorithm brings over just looking at the multi-block problem as a single min-max bilevel optimization problem and solving it thusly.\
> **Response**: Yes, the considered problem can be viewed as standard min-max bilevel optimization problem  with a single block.  The key benefit of viewing it as multi-block problems instead of single min-max bilevel optimization is that our algorithm only needs to sample and process the data for the sampled few blocks (i.e., tasks). For problems with many tasks (e.g., ogbg-molpcba with 128 tasks), it is expensive to sample and process a mini-batch of data points for all 128 tasks if we want to maintain a certain batch size for each task to reduce the variance of its stochastic gradient. In some scenarios (e.g., online advertising), it might be the case that only data for a few tasks are available at each iteration, which makes the algorithm that requires data for all tasks at each iteration not applicable. Hence, our algorithm is more efficient and flexible.
>
> **References**:
>
> [8] Alex Gu, Songtao Lu, Parikshit Ram, and Lily Weng. Min-max bilevel multi-objective optimization with applications in machine learning, 2022.
>
> [38] Zhuoning Yuan, Yan Yan, Milan Sonka, and Tianbao Yang. Robust deep auc maximization: A new surrogate loss and empirical studies on medical image classification. arXiv preprint arXiv:2012.03173, 2020.

---

### Official Review · Reviewer_6MPu · 2022-07-12

**Rating:** 7
**Confidence:** 3
**Soundness:** 3 good
**Presentation:** 3 good
**Contribution:** 2 fair

**Summary:**

This paper studies the problem of multi-block min-max bilevel optimization, where the dual variables and the associated lower problem are divided into blocks. To reduce the computational expense, this paper proposes to sample a batch of blocks at each iteration and perform update only for those blocks. In theory, the sample complexity of the proposed algorithm matches the optimal order for stochastic nonconvex optimization. The proposed method is then applied to multi-task deep (partial) AUC maximization, which can be formulated as a multi-block min-max bilevel optimization problem. Extensive experiments also verify the effectiveness of the proposed method in this scenario.

**Questions:**

This paper have briefly compared the use of block sampling with previous work [27]. However, I wonder whether it will bring anything new to the algorithm design or theoretical analysis, or it is just a direct application.

Typo: line 254, sigmod -> sigmoid

**Limitations:**

Limitations of this work have been properly discussed.

**Strengths And Weaknesses:**

Strength:

\+ This paper provides an interesting perspective of viewing the multi-task AUC maximization as a multi-block bilevel optimization problem. Such a generalization may inspire future research on this important area.

\+ This paper is technically sound, supported by non-trivial theoretical analysis and extensive experiments.

Weakness:

\- The block sampling and moving averaging gradient estimator have some overlap with previous work, which somehow affects the novelty of the algorithm design.

---

> ### Author Response · Authors · 2022-07-29
> **Regarding Reviewer 6MPu’s Concerns**
>
> Thank you for your positive rating! Below we address your question.
>
> **Q**: This paper have briefly compared the use of block sampling with previous work [27]. However, I wonder whether it will bring anything new to the algorithm design or theoretical analysis, or it is just a direct application.\
> **Response**: We do not consider this work as a direct application of [27] though block sampling is used in both papers. In particular, our problem has an additional maximization structure with multiple blocks, which is not present in [27]. Hence, the algorithmic design for handling the maximization problem and its synthesis with other updates requires non-trivial work. In particular, we have adopted the simplest block-wise stochastic gradient ascent update for handling the maximization problems and managed to prove that it does not worsen the complexity compared with [27].
>
> Thank you for pointing out the typo. We  have fixed it.
>
> **References**:
>
> [27] Zi-Hao Qiu, Quanqi Hu, Yongjian Zhong, Lijun Zhang, and Tianbao Yang. Large-scale stochastic optimization of ndcg surrogates for deep learning with provable convergence, 2022.

---

### Official Review · Reviewer_h3Nq · 2022-07-16

**Rating:** 4
**Confidence:** 4
**Soundness:** 2 fair
**Presentation:** 3 good
**Contribution:** 2 fair

**Summary:**

In this paper, the authors study a multi-block min-max optimization problem and propose a single-loop randomized stochastic algorithm to solve this class of problems with provable convergence rate guarantees. It is shown that the achieved convergence rate is in the same order as solving stochastic nonconvex optimization problems. Multiple numerical results show the effectiveness of the method on multi-task AUC problems.

**Questions:**

Besides the above comments, there are also several questions:

1, it seems that the main motivation for having this model was from [37]. Why is the lower-level problem importantly incorporated in this model?

2, the paper focuses on the multi-block model. Why does only $\alpha$ have multi-blocks? how about $x$?

3, the algorithm uses the momentum term. Will this should give a faster convergence rate？ or which step (min, max, or lower-level problem) restricts the final convergence rate?

4, In theorem 2.7, if $\eta_1$ and $\eta_2$ are chosen according to lemma 2.4 and lemma 2.5, then the algorithm should not converge to an $\epsilon$-stationary point unless they are decreasing or proportional $\epsilon$? (because there is a $T$ in front of  $\eta_1$ and $\eta_2$).

5, I am confused that how the lower-level problem is formulated between lines 258 and 259.

6, is there an optimal solution for the lower level optimization problem because it is just an unconstrained quadratic problem?

7, the authors claimed that $1/\epsilon^4$ is the optimal complexity for nonconvex optimization under a general unbiased stochastic oracle model. Which reference showed the lower bound in this case so that we can claim this is the optimal complexity? To my understanding, variance reduction or accelerated technique, in general, can have a better complexity, especially I suppose that the authors should clarify the conditions under which the algorithm can achieve the optimal complexity.



**Limitations:**

Yes.

**Strengths And Weaknesses:**

 The strengths of this paper:

1, this work considers a very general bilevel optimization problem, which covers a wide range of machine learning problems as special cases.

2, a stochastic multi-block minmax bilevel optimization algorithm is proposed, which takes the block structure of the problem, stochasticity of data samples, and variance reduction into account.

3, this paper also provides the theoretical convergence rate analysis of the proposed algorithm and shows that the achieved rate matches the optimal complexity of nonconvex optimization under a general unbiased stochastic oracle model.

The weaknesses of this work are

1,  (originality) the problem seems a combination of minmax optimization and bilevel optimization, especially under the conditions that the subproblem w.r.t. maximization is strongly convex and the lower-level problem is also strongly convex. How to position the novelty of this problem and algorithms among the existing work is not clear.

2, (quality) The major contribution is the theoretical convergence rate analysis, but what are the challenges of showing this rate was not discussed technically

---

> ### Author Response · Authors · 2022-07-29
> **Regarding Reviewer h3Nq’s Questions**
>
> Thank you for your useful comments! Below we address your questions.
>
> **Q1**: It seems that the main motivation for having this model was from [37]. Why is the lower-level problem importantly incorporated in this model?\
> **Response**: This paper focuses on developing efficient optimization algorithms for a family of multi-block min-max bilevel optimization problems. The lower-level problem in our bilevel reformulation is **equivalent** to the inner gradient step of the compositional formulation in [37]. This bilevel formulation enables us to use the proposed algorithm to tackle **multi-task** deep AUC maximization, which was not considered in [37].  We do not introduce any new modeling approach except that we tackle the challenges of solving multi-task deep AUC maximization. \
> Besides, our algorithm is applicable to a new formulation of multi-task deep **partial AUC** maximization problem described in section 3.2, which has nothing to do with [37]. In this application, the lower-level problem is important to tackle the challenge of selecting top ranked negative examples, which is arguably better than existing approaches for partial AUC maximization [40] that introduce many more auxiliary variables for optimization.
>
> **Q2**: The paper focuses on the multi-block model. Why does only $\alpha$ have multi-blocks? how about $x$ ?\
> **Response**: In this paper, we  do not consider multi-block structure in the model parameter $\textbf{x}$. The multi-block is not referred to the model in this paper (we never used the term ``multi-block model" in the paper). Instead, it refers to the objective structure, which mainly comes from the multi-task structure in applications.  This multi-block perspective allows us to only sample and process data for a few sampled tasks per-iteration instead of sampling/processing data for all tasks at each iteration, which is prohibitive when the number of tasks is large. Although each task has its own model parameters (e.g., task-specific classifier head), our algorithm updates all model parameters (including shared encoder network and task-specific classifier heads) at each iteration by using the updated stochastic gradient estimator $\mathbf z_{t+1}$.
>
> **Q3**:  The algorithm uses the momentum term. Will this should give a faster convergence rate? or which step (min, max, or lower-level problem) restricts the final convergence rate?\
> **Response**: The momentum term has two benefits: (i) it yields an optimal complexity in the order of $O(1/\epsilon^4)$ under our considered assumption according to Arjevani et al. [1]; (ii) it does not require an extremely large batch size as in previous works without using the momentum term. For example, the stochastic algorithm considered in Lin et al. [23] for min-max problems does not use momentum term but requires a large batch size in the order of $O(1/\epsilon^2)$ for achieving the same complexity. The stochastic algorithm proposed in Hong et al. [13] for bilevel problems does not use momentum term but suffers a worse complexity in the order of $O(1/\epsilon^5)$. A recent work Chen et al. achieves complexity of $O(1/\epsilon^4)$, with a double loop SGD method, i.e., the update for the lower-level variable $y$ requires an inner-loop with multiple iterations. Using the momentum for estimating the gradient of $\mathbf x$ is helpful in reducing its error due to the inexact dual variables and lower-level solutions.
>
>
> **Q4**:  In theorem 2.7, if $\eta_1$ and $\eta_2$ are chosen according to lemma 2.4 and lemma 2.5, then the algorithm should not converge to an $\epsilon$-stationary point unless they are decreasing or proportional $\epsilon$? (because there is a T in front of $\eta_1$ and $\eta_2$).\
> **Response**: You are right. $\eta_1,\eta_2$ are  proportional to $\epsilon^2$, i.e. $\eta_1,\eta_2=\mathcal{O}\left(|\mathcal{B}_i^t|\epsilon^2\right)$. We have stated this clearly in Theorem 2.7 (cf. line 201).
>
>
> **Q5**: How is the lower-level problem formulated between lines 258 and 259?\
> **Response**: Such formulation has been used in [27]. The lower-level problem is formulated so that its solution $\lambda_k(\textbf{w})$ serves as a threshold of the top K prediction scores. A detailed proof can be found in Lemma 5.1 in [27].
>
>
> **Q6**: Is there an optimal solution for the lower level optimization problem because it is just an unconstrained quadratic problem?\
> **Response**: The answer is yes for the multi-task deep AUC maximization problem. However, it is not efficient to compute this optimal solution because it depends on all data for each task. We are not able to compute this optimal solution by passing through all data points at each iteration.

---

> > ### Comment · Reviewer_h3Nq · 2022-08-09
> > **need more clarification on the contribution of this work**
> >
> > Thanks for your response.
> >
> > 1, Q3: it seems that there are existing works that can achieve faster convergence rate by using the momentum technique, e.g.,
> >
> > Yang et al, Provably faster algorithms for bilevel optimization, 2021.
> > Li et al, Local stochastic bilevel optimization with momentum-based variance reduction, 2022.
> > Khanduri et al, A near-optimal algorithm for stochastic bilevel optimization via double-momentum, 2021
> >
> > why does not this algorithm can have a faster rate?
> >
> > 2, Q6: it is dependent on all the data. But the issue here is that even running gradient descent is also using stochastic data points. So, why does not use the optimal closed form solution based on the sampled points to update the lower level optimization variables?

---

> > > ### Author Response · Authors · 2022-08-09
> > > **Clarifications are given below. We believe they are well clarified. Please let us know if there are more questions.**
> > >
> > > **Q1: why not faster rate**.
> > >
> > > **A**: There are two reasons.
> > >
> > > First, we do **not** make the assumption that the stochastic gradient is Lipchitz continuous or on-average Lipschitz continuous. This is the key condition used in existing works based on variance reduction (e.g., STORM) to derive a faster rate of $O(1/\epsilon^3)$ for finding an $\epsilon$-level stationary point.  For example, in Yang et al., their Assumption 2 (b) (c) (d) make the assumption that stochastic gradients and Hessians are Lipschitz continuous. Khanduri et al. makes similar assumptions in their proof (though not explicitly mentioned). Please check their proof here (https://openreview.net/attachment?id=HjFtRc83eBB&name=supplementary_material), in page 22 when deriving (b) from (a) they use the Lipschitz condition of the stochastic gradient $\nabla_y f(x, y, \xi)$ and stochastic Jacobian $\nabla_{xy}g(x, y, \zeta^0)$. in Li et al., the authors use similar assumptions. In their proof (https://arxiv.org/pdf/2205.01608.pdf), in page 32 from inequality (c) to inequality (d), they use the on-average Lipschitz continuity of the stochastic gradient (which is not made explicit in their main content). Indeed, the assumption that stochastic gradient is Lipchitz continuous or on-average Lipschitz continuous is necessary for deriving a faster rate in the order $O(1/\epsilon^3)$, which is consistent with the lower bound in Arjevani et al. (2019) (which is referred to as mean-squared smoothness property in their paper).  In contrast, in our proof of the main Theorem 2.7, we never use the condition that stochastic gradient is Lipchitz continuous or on-average Lipschitz continuous.
> > >
> > > Second, the task sampling makes it more challenging to use variance reduction to derive a faster rate. The reason is that there is not only error in the stochastic gradient for each task but also there is noise caused by the sampling of tasks. Directly using the variance reduction technique (e.g. STORM)  for the sampled task does not necessarily yield a faster rate. Please check this recent study (https://arxiv.org/abs/2207.08540) for illustration on a compositional problem. The authors proposed another variance reduced estimator to track a function mapping with multiple blocks for achieving a faster rate.   We do not want to overwhelm the reviewer with heavy mathematics to illustrate this issue here. Please refer to https://arxiv.org/abs/2207.08540 and note that our work is a concurrent work. It is possible to utilize their variance reduced estimator for derive a faster rate for our problem but is out of scope of the current work.
> > >
> > > We will remark the difference from existing works and highlight the challenges in the final version.
> > >
> > > Yossi Arjevani, Yair Carmon, John C Duchi, Dylan J Foster, Nathan Srebro, and Blake Woodworth. Lower bounds for non-convex stochastic optimization. arXiv preprint arXiv:1912.02365, 2019.
> > >
> > > Zhishuai Guo, Quanqi Hu, Lijun Zhang, and Tianbao Yang. Randomized stochastic variance-reduced methods for multi-task stochastic bilevel optimization.
> > >
> > >
> > > **Q2 why does not use the optimal closed form solution based on the sampled points to update the lower level optimization variables?**
> > >
> > > **A**: This would not work at least it does not give a satisfactory result. The reason is this. In this case, let us view the problem from the compositional optimization perspective, i.e., $f(g(w))$, where $g(w)$ is equivalent to the optimal solution of a bilevel problem, i.e.,  $u^* = \arg\min_{u}|u - g(w)|^2$. If we simply use  stochastic samples to compute a $u=\hat g(w)$ in place of $u^*$ for computing a gradient estimator $\nabla f(u)\nabla \hat g(w)$. This corresponds to the biased stochastic gradient descent (BSGD) method analyzed in Hu et al.  (2020). Their theoretical result shows that the optimization error will depend on the batch size in computing $\hat g(w)$, i.e., the smaller the batch size, the larger the optimization error. Unless it uses huge batch size in the order of $1/\epsilon^2$, the BSGD algorithm does not converge. Our update here for the lower level variable $u$ uses stochastic gradient descent update, which can leverage the strong convexity of the lower level problem to enjoy diminishing error of the lower level update in the long term with respect to their optimal solutions (i.e., Lemma 2.4 in the paper).
> > >
> > > Hu et al. Biased stochastic first-order methods for conditional stochastic optimization and applications in meta learning. Advances in Neural Information Processing Systems, 33, 2020.

---

> ### Author Response · Authors · 2022-07-29
> **Regarding Reviewer h3Nq’s Questions (Cont’d)**
>
> **Q7**: The authors claimed that $1/\epsilon^4$ is the optimal complexity for nonconvex optimization under a general unbiased stochastic oracle model. Which reference showed the lower bound in this case so that we can claim this is the optimal complexity? To my understanding, variance reduction or accelerated technique, in general, can have a better complexity, especially I suppose that the authors should clarify the conditions under which the algorithm can achieve the optimal complexity.\
> **Response**: It has been proved in Arjevani et al. [1] that for smooth potentially non-convex minimization problem with an access of unbiased stochastic gradient oracle under **a bounded variance condition**, any algorithm requires at least $1/\epsilon^4$ queries to find an $\epsilon$-stationary point. Since our problem includes smooth non-convex minimization as a special case, hence $1/\epsilon^4$ is also the optimal complexity for our problem under the bounded variance condition.  It is possible to improve the complexity to $1/\epsilon^3$ by using variance reduction techniques, however, one requires additional **Lipschitz assumption** on the stochastic gradient. In our work, since we only assume bounded variance condition, the complexity $1/\epsilon^4$ we achieve is optimal in terms of $\epsilon$.  We have made it more clear in the revision (please see lines 89 - line 91 in the revision).
>
>
>
> **References**:
>
> [1] Yossi Arjevani, Yair Carmon, John C Duchi, Dylan J Foster, Nathan Srebro, and Blake Woodworth. Lower bounds for non-convex stochastic optimization. arXiv preprint arXiv:1912.02365, 2019.
>
> [13] Mingyi Hong, Hoi-To Wai, Zhaoran Wang, and Zhuoran Yang. A two-timescale framework for bilevel optimization: Complexity analysis and application to actor-critic. arXiv preprint arXiv:2007.05170, 2020.
>
> [15] Feihu Huang, Shangqian Gao, Jian Pei, and Heng Huang. Accelerated zeroth-order momentum methods from mini to minimax optimization. arXiv preprint arXiv:2008.08170, 2020.
>
> [23] Tianyi Lin, Chi Jin, and Michael I Jordan. On gradient descent ascent for nonconvex-concave minimax problems. arXiv preprint arXiv:1906.00331, 2019.
>
> [27] Zi-Hao Qiu, Quanqi Hu, Yongjian Zhong, Lijun Zhang, and Tianbao Yang. Large-scale stochastic optimization of ndcg surrogates for deep learning with provable convergence, 2022.
>
> [37] Zhuoning Yuan, Zhishuai Guo, Nitesh Chawla, and Tianbao Yang. Compositional training for end-to-end deep AUC maximization. In International Conference on Learning Representations, 2022.
>
> [40] Dixian Zhu, Gang Li, Bokun Wang, Xiaodong Wu, and Tianbao Yang. When AUC meets DRO: optimizing partial AUC for deep learning with non-convex convergence guarantee. CoRR, abs/2203.00176, 2022.
>
> Tianyi Chen, Yuejiao Sun and Wotao Yin. Tighter analysis of alternating stochastic gradient method for stochastic nested problems. In Proc. of Advances in Neural Information Processing Systems, 2021.

---

> > ### Comment · Reviewer_h3Nq · 2022-08-09
> > **Lipschitz conditions**
> >
> > Thanks for your response.
> > Is it possible to specify what is exactly the Lipschitz continuous assumption required? Please note that the Lipschitz continuity on function $f$ is needed, which is strong. Will this affect the lower bound?

---

> > > ### Author Response · Authors · 2022-08-09
> > > **Thank you!**
> > >
> > > The Lipschitz continuous assumption mentioned in our response earlier for deriving a faster rate refers to the Lipschitz continuous condition of the stochastic gradient, stochastic Jacobian and stochastic Hessian. We give an example of Yang et al, e.g., $|\nabla F(x, y, \xi) - \nabla F(x', y', \xi)|\leq L\|(x, y) - (x', y')|$, where $\xi$ denotes a stochastic sample. The same conditions are assumed for stochastic Jacobian and stochastic Hessian of the lower level problem.
> > >
> > > In our work, we assume the weaker Lipschitz continuous assumption, i.e., Lipschitz continuity on the full gradient, full Jocobian and full Hessian (assumption 2.2 point 2 and point 4). We also assume the upper function $f$ is Lipschitz continuous in terms of $y$, which is also a standard assumption in bilevel optimization (please check Yang et al., Li et al., Ghadimi and Mengdi Wang.) In addition, we assume  $f$ is Lipschitz continuous in terms of $x$ as well (which is also assumed in Yang et al. and Li et al.). In our paper, this is due to that we have an additional challenge of task sampling.  For the single task setting, the Lipschitz continuity of $f$  in terms of $x$ can be removed, which will be consistent with the condition made in lower bound analysis (Arjevani et al.).  It is unclear (possible) whether the Lipschitz continuity of $f$  in terms of $x$ will affect the lower bound. We have mad our statement more precise in the revision, i.e., "Our result for the singl-block setting matches the lower bound for solving smooth, potentially nonconvex optimization through queries to an unbiased stochastic gradient oracle under a bounded variance condition \cite{2019lowerarjevani}". We also add a remark under Assumption 2.2 regarding the Lipschitz continuity assumption in the revision.
> > >
> > > Yang et al, Provably faster algorithms for bilevel optimization, 2021.
> > >
> > > Li et al, Local stochastic bilevel optimization with momentum-based variance reduction, 2022.
> > >
> > > Saeed Ghadimi and Mengdi Wang. Approximation methods for bilevel programming. arXiv
> > > preprint arXiv:1802.02246, 2018
> > >
> > > Yossi Arjevani, Yair Carmon, John C Duchi, Dylan J Foster, Nathan Srebro, and Blake Woodworth. Lower bounds for non-convex stochastic optimization. arXiv preprint arXiv:1912.02365, 2019.

---

> ### Author Response · Authors · 2022-08-02
> **Regarding Reviewer h3Nq’s Concerns about weakness and contributions**
>
> **Q**:  (Originality) How to position the novelty of this problem and algorithms among the existing work is not clear.  (Quality) what are the challenges of showing this rate was not discussed technically.
>
> **A**:  The novelty of this problem is that there are **multiple lower level** problems and **multiple blocks** of dual variables for maximization. The challenge is how to tackle **a large number** of blocks efficiently. As far as we are concerned, there is no work that tackles this challenge in the minmax bilevel setting. Most existing works for minmax optimization or bilevel optimization assume there is only single block except for [27], which considers a bilevel optimization problem with multiple blocks.
>
> We do not consider this work as a direct application of [27] though block sampling is used in both papers. In particular, our problem has an additional maximization structure with multiple blocks, which is not present in [27]. Hence, the algorithmic design for handling the maximization problem and its synthesis with the SGD update for the sampled lower level problems and momentum-based updates for the primal variable requires non-trivial work. In particular, we have adopted the simplest block-wise stochastic gradient ascent update for handling the maximization problems and managed to prove that it does not worsen the complexity compared with [27]. Please also check the response to your Q3 below.
>
> **Novelty for handling the maximization problems for deriving $O(1/\epsilon^4)$ complexity**. We do not use momentum update on the dual variables as in previous works, e.g, [15], Qiu et al. (2020), as it will complicate the algorithm with more parameter for tuning. The reason is that we bound the error of the gradient estimator $\mathbf z_{t+1}$ with respect to $\nabla F(\mathbf x_t)$ differently (please refer to proof of Lemma A.1). In particular, we do not introduce the term $\|\alpha_{t+1} - \alpha_t\|^2$  (e.g., Lemma B.4 in Qiu et al. 2020, and Lemma 41 in [15]). To tackle the additional term $\|\alpha_{t+1} - \alpha_t\|^2$, both [15] and Qiu et al. use a momentum update for the dual variable, which is to cancel the additional $\|\alpha_{t+1} - \alpha_t\|^2$ mentioned above. Otherwise it will yield worse complexity. The main difference is that we do NOT establish the recursion for $\|\mathbf z_{t+1} - \nabla F(\mathbf x_t, \alpha_t, \mathbf y_t)\|^2$ as in the prior works. Instead, we directly build the recursion for $\|\mathbf z_{t+1} - \nabla F(\mathbf x_t)\|^2$.
>
> We will add some discussion in the proof of the revision to clarify the differences and our novelty.
>
> Qiu et al. (2020) Single-Timescale Stochastic Nonconvex-Concave Optimization for  Smooth Nonlinear TD Learning.

---

> > ### Comment · Reviewer_h3Nq · 2022-08-09
> > **weakness of the theorem**
> >
> > Thanks for your response. Note that this algorithmic design may have a reasonable good convergence rate, but the Lipschitz continuity of the lower level function is assumed, which is a very strong assumption. You have this assumption due to the use of the moving average recursion. Therefore, it is hard to compare the achievable rate to the existing ones.

---

> > > ### Author Response · Authors · 2022-08-09
> > > **It is a typo not a weakness.**
> > >
> > > We are sorry for the confusion. Actually the Lipschitz continuity of the lower level function is **not** used in our proof (please check the detailed proof of Theorem 2.7 in supplement). It was added accidentally. We have removed this assumption in the revision.

---

> ### Author Response · Authors · 2022-08-06
> **Does our rebuttal address your concerns?**
>
> Dear reviewer h3Nq,
>
> Please help check our rebuttal and let us know if you have additional questions or concerns about our paper and responses.
>
> Thank you!
> Authors

---

### Meta-Review · Area_Chair_Dydm · 2022-08-27

**Recommendation:** Accept
**Confidence:** Certain

**Metareview:**

This paper propose novel algorithm for a class of minimax problems. The iteration complexity is established. The proposed algorithm is applied to AUC maximization -- a very important problem in machine learning. Considering the contributions in both theory and practice, this is a solid work to the machine learning community.

**Award:**

No

---

### Decision · Program_Chairs · 2022-09-14

Accept